# Iron derived from autophagy-mediated ferritin degradation induces cardiomyocyte death and heart failure in mice

Jumpei Ito[1,2], Shigemiki Omiya[1], Mara-Camelia Rusu[1], Hiromichi Ueda[3], Tomokazu Murakawa[1], Yohei Tanada[1], Hajime Abe[1], Kazuki Nakahara[1], Michio Asahi[2], Manabu Taneike[1,3], Kazuhiko Nishida[1], Ajay M Shah[1], Kinya Otsu[1]*

[1]The School of Cardiovascular Medicine and Sciences, King's College London British Heart Foundation Centre of Excellence, London, United Kingdom; [2]Department of Pharmacology, Faculty of Medicine, Osaka Medical College, Osaka, Japan; [3]Department of Cardiovascular Medicine, Graduate School of Medicine, Osaka University, Osaka, Japan

**Abstract** Heart failure is a major public health problem, and abnormal iron metabolism is common in patients with heart failure. Although iron is necessary for metabolic homeostasis, it induces a programmed necrosis. Iron release from ferritin storage is through nuclear receptor coactivator 4 (NCOA4)-mediated autophagic degradation, known as ferritinophagy. However, the role of ferritinophagy in the stressed heart remains unclear. Deletion of *Ncoa4* in mouse hearts reduced left ventricular chamber size and improved cardiac function along with the attenuation of the upregulation of ferritinophagy-mediated ferritin degradation 4 weeks after pressure overload. Free ferrous iron overload and increased lipid peroxidation were suppressed in NCOA4-deficient hearts. A potent inhibitor of lipid peroxidation, ferrostatin-1, significantly mitigated the development of pressure overload-induced dilated cardiomyopathy in wild-type mice. Thus, the activation of ferritinophagy results in the development of heart failure, whereas inhibition of this process protects the heart against hemodynamic stress.

*For correspondence:
kinya.otsu@kcl.ac.uk

Competing interests: The authors declare that no competing interests exist.

## Introduction

Heart failure is the leading cause of death in developed countries (*Ponikowski et al., 2016*). There is substantial evidence to suggest the involvement of oxidative stress and cardiomyocyte death in the pathogenesis of heart failure (*Whelan et al., 2010*). Iron metabolism in heart failure patients is dysregulated (*Lavoie, 2020*), but it remains unclear whether these changes are pathogenetic and detrimental or adaptive and protective for the heart. While iron is essential for oxidative phosphorylation, metabolite synthesis, and oxygen transport (*Andrews and Schmidt, 2007*), it can generate toxic reactive hydroxyl radicals through the Fenton reaction (*Papanikolaou and Pantopoulos, 2005*). Iron-dependent necrotic cell death is characterized by iron overload and an increased level of lipid reactive oxygen species (ROS) such as lipid hydroperoxides, leading to phospholipid damage, plasma membrane disruption, and caspase- and necrosome-independent cell death (*Dixon et al., 2012*). Ferroptosis is a form of iron-dependent necrosis. Multiple molecular components contribute to the execution of ferroptosis, such as transferrin–iron, cystine–glutathione, and glutamine pathways (*Gao et al., 2015*). Glutathione peroxidase 4 (GPX4) is a phospholipid hydroperoxide-reducing enzyme that uses glutathione as a substrate. The failure of GPX4 to clear lipid ROS leads to lipid peroxidation and ferroptosis (*Yang et al., 2014*). Glutamate through the glutamine-

fueled intracellular metabolic process glutaminolysis induces ferroptosis (*Gao et al., 2015*). Ferroptosis has been implicated in the pathological process associated with ROS-induced tissue injury, such as ischemia/reperfusion in the brain, kidney, and heart (*Fang et al., 2019*; *Linkermann et al., 2014*; *Tuo et al., 2017*).

Iron is stored in ferritin protein complexes in the cell to prevent an increase in the size of the labile iron pool that normally follows iron overload. Ferritin is a ubiquitously expressed cytosolic heteropolymer composed of H-chains (FTH1) and L-chains (FTL) (*Arosio et al., 2009*). FTH1 has ferroxidase activity and sequestrates ferrous iron ($Fe^2$) from the Fenton reaction in which the spontaneous oxidation to ferric iron ($Fe^3$) donates single electrons to transform innocuous hydrogen peroxide to highly toxic hydroxyl free radicals. In the case of iron overload, ferritin subunits are induced by inactivating the iron regulatory protein (IREB)/iron-responsive element pathway. By contrast, under conditions of iron deficiency or increased iron requirement, ferritin is degraded and mediated through a selective form of autophagy, called ferritinophagy. Nuclear receptor coactivator 4 (NCOA4) is a cargo receptor for ferritinophagy that interacts with FTH1 and promotes the transport of ferritin to the autophagosome for degradation (*Dowdle et al., 2014*; *Mancias et al., 2014*). NCOA4-dependent iron release from ferritin storage is necessary for erythropoiesis (*Bellelli et al., 2016*).

## Results

### Generation and characterization of cardiomyocyte-specific NCOA4-deficient mice

To examine the in vivo role of NCOA4-dependent ferritin degradation in the heart, cardiomyocyte-specific NCOA4-deficient mice were generated (*Figure 1—figure supplement 1A and B*). The homozygous floxed *Ncoa4* mice (*Ncoa4*flox/flox) appeared normal and were externally indistinguishable from littermates of other genotypes. The *Ncoa4*flox/flox mice were crossed with transgenic mice expressing α-myosin heavy chain (*Myh6*) promoter-driven Cre recombinase (Myh6-Cre) (*Nishida et al., 2004*) to generate cardiomyocyte-specific NCOA4-deficient mice (*Ncoa4*−/−), *Ncoa4*flox/flox;Myh6-Cre+. The *Ncoa4*flox/flox;Myh6-Cre− littermates were used as controls (*Ncoa4*+/+). The *Ncoa4*+/+ and *Ncoa4*−/− mice were born at the expected Mendelian ratio (54 and 51 mice, respectively), and they grew to adulthood and were fertile. The protein and mRNA levels of NCOA4 were significantly decreased in *Ncoa4*−/− hearts by 84% and 81% compared to control, respectively (*Figure 1—figure supplement 1C and D*). No differences in any physiological or echocardiographic parameters were observed between the *Ncoa4*−/− and *Ncoa4*+/+ mice (*Figure 1—source data 2*).

### Attenuation of pressure overload-induced cardiac remodeling in NCOA4-deficient mice

To examine whether NCOA4 is related to cardiac remodeling in vivo, *Ncoa4*+/+ and *Ncoa4*−/− mice were subjected to pressure overload employing transverse aortic constriction (TAC) and evaluated 4 weeks after the operation. Pressure overload increased the left ventricular (LV) chamber size, indicated by the end-diastolic and end-systolic LV internal dimensions, and reduced fractional shortening (an index of contractility) in *Ncoa4*+/+ mice compared to sham-operated controls (*Figure 1A and B*). These pressure overload-induced changes in heart size and function were suppressed in *Ncoa4*−/− mice. The calculated LV mass, LV weight-to-tibia length ratio, and the cross-sectional area of cardiomyocytes, which are parameters for cardiac hypertrophy, were elevated by pressure overload in both *Ncoa4*+/+ and *Ncoa4*−/− mice, but those were significantly lower in TAC-operated *Ncoa4*−/− mice than in TAC-operated controls (*Figure 1B,C and D*). TAC-operated *Ncoa4*+/+ mice displayed higher mRNA expression levels of the cardiac remodeling markers, *Nppa*, *Nppb*, and *Myh7* than TAC-operated *Ncoa4*−/− mice (*Figure 1—figure supplement 2*). Furthermore, the lung weight-to-tibia length ratio, an index of lung congestion, was significantly elevated in TAC-operated *Ncoa4*+/+ mice compared to both sham-operated *Ncoa4*+/+ and TAC-operated *Ncoa4*−/− mice (*Figure 1C*). The extent of pressure overload-induced fibrosis in heart sections and the mRNA levels of *Col1a2* and *Col3a1*, markers for fibrosis, in *Ncoa4*−/− mice were lower than in *Ncoa4*+/+ mice (*Figure 1D* and *Figure 1—figure supplement 2*). There were no differences in echocardiographic parameters between TAC-operated Myh6-Cre+ and Myh6-Cre− mice 4 weeks after TAC (*Figure 1—figure supplement 3*). Thus, the overexpression of Cre recombinase in the heart has no effect on

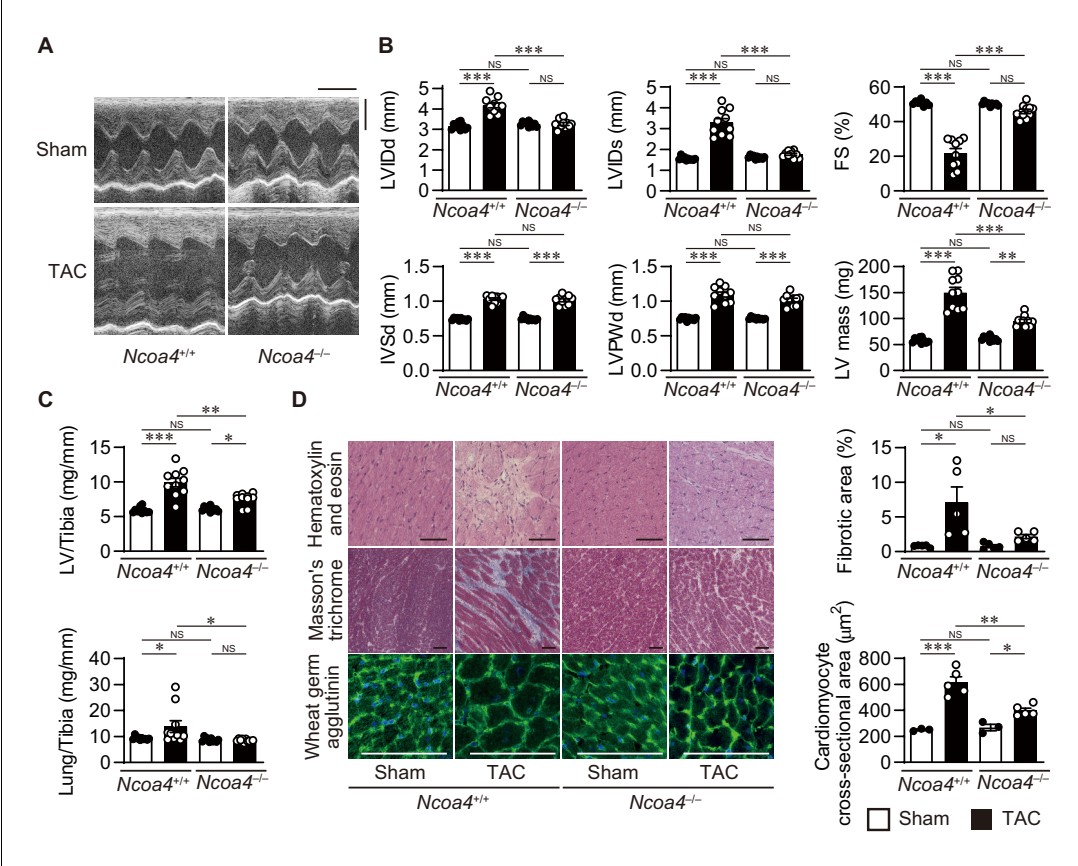

**Figure 1.** Cardiomyocyte-specific *Ncoa4* ablation attenuated the development of pressure overload-induced heart failure. The *Ncoa4*[+/+] and *Ncoa4*[−/−] mice were subjected to pressure overload by transverse aortic constriction (TAC) and analyzed 4 weeks after the operation. (**A**) Representative images of transthoracic M-mode echocardiographic tracing. Scale bars, 0.1 s and 2 mm, respectively. (**B**) Echocardiographic parameters of the mice (*n* = 10 biologically independent samples). LVIDd and LVIDs, end-diastolic and end-systolic left ventricular (LV) internal dimensions; IVSd, end-diastolic interventricular septum thickness; LVPWd, end-diastolic LV posterior wall thickness; FS, fractional shortening. (**C**) Physiological parameters of the mice (*n* = 10 biologically independent samples). (**D**) Representative images of the hematoxylin-eosin-stained (upper), Masson's trichrome-stained (middle), and wheat germ agglutinin-stained (lower) heart sections. Scale bar, 50 μm. The upper and lower right graphs show the ratio of the fibrotic area to whole heart section and the cross-sectional area of cardiomyocytes, respectively (*n* = 5 biologically independent samples). The data were evaluated by one-way analysis of variance (ANOVA), followed by Tukey–Kramer's post hoc test. *p<0.05, **p<0.001, ***p<0.0001. NS, p>0.05. Exact p-values are provided in *Supplementary file 1*.

The online version of this article includes the following source data and figure supplement(s) for figure 1:

**Source data 1.** Source data for *Figure 1*.
**Source data 2.** Physiological and echocardiographic parameters in 8- to 10-week-old *Ncoa4*[+/+] and *Ncoa4*[−/−] mice at baseline.
**Source data 3.** Source data for data table provided in *Figure 1—source data 2*.
**Figure supplement 1.** Generation of cardiomyocyte-specific nuclear receptor coactivator 4 (NCOA4)-deficient mice.
**Figure supplement 1—source data 1.** Source data for *Figure 1—figure supplement 1*.
**Figure supplement 2.** Cardiac remodeling markers in transverse aortic constriction (TAC)-operated *Ncoa4*[−/−] mice.
**Figure supplement 2—source data 1.** Source data for *Figure 1—figure supplement 2*.
**Figure supplement 3.** Myh6-Cre transgene does not alter the heart response to pressure overload stress.
**Figure supplement 3—source data 1.** Source data for *Figure 1—figure supplement 3*.

pressure overload-induced cardiac remodeling. Taken together, NCOA4 deficiency attenuated pressure overload-induced cardiac remodeling, including cardiac hypertrophy and dysfunction, chamber dilation, and fibrosis.

# Attenuation of upregulation of ferritinophagy in pressure-overloaded NCOA4-deficient hearts

The level of ferritinophagy in pressure-overloaded hearts was then evaluated 4 weeks after TAC. The protein level of FTH1 was decreased in $Ncoa4^{+/+}$ hearts compared to both sham-operated $Ncoa4^{+/+}$

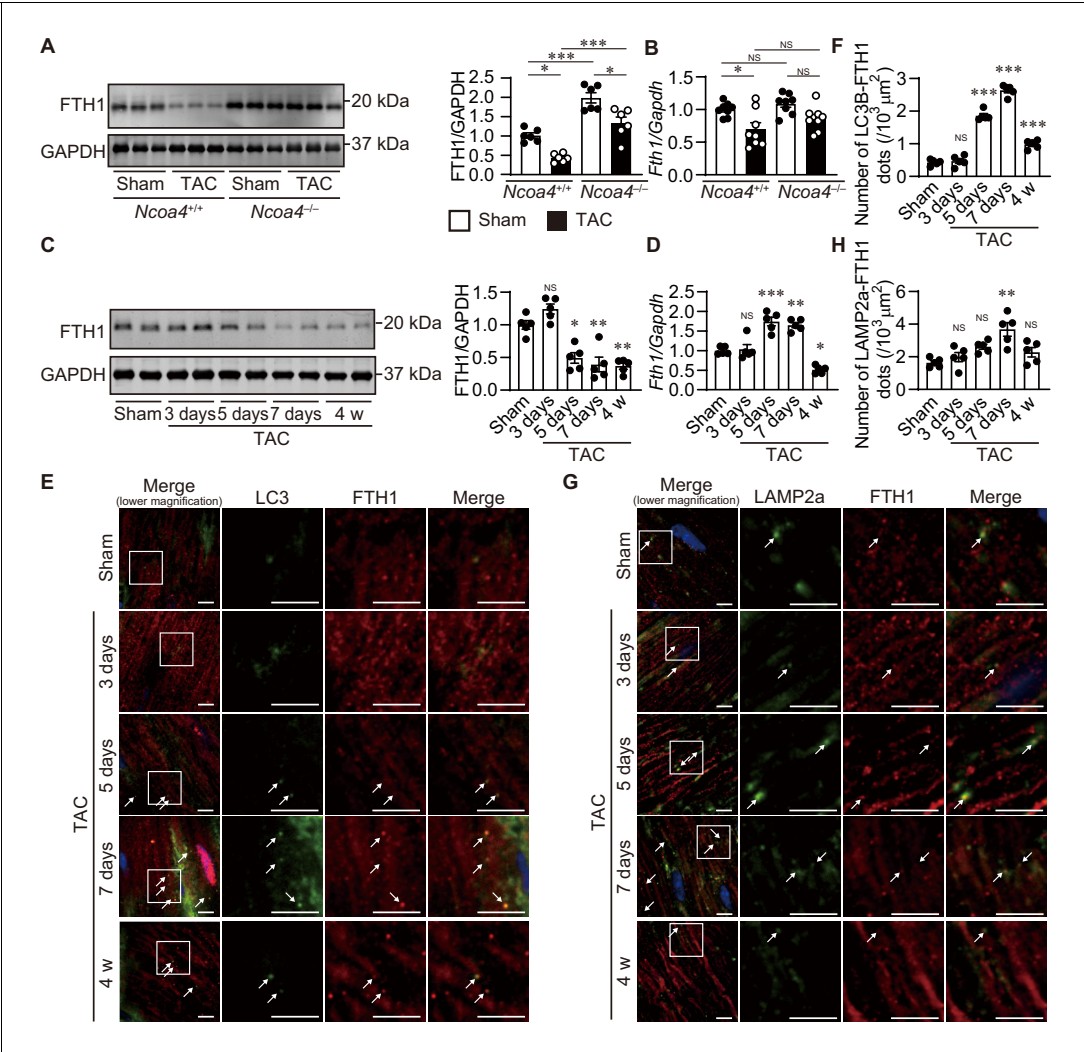

**Figure 2.** The time course of ferritinophagy in the heart after transverse aortic constriction (TAC). (A) Western blot analysis of FTH1 in $Ncoa4^{+/+}$ and $Ncoa4^{-/-}$ hearts 4 weeks after TAC. GAPDH was used as the loading control. The right-hand graphs show the densitometric analysis. The average value for sham-operated $Ncoa4^{+/+}$ hearts was set to 1 (biologically independent samples: $n = 6$). (B) mRNA expression of $Fth1$ in the heart 4 weeks after TAC. $Gapdh$ mRNA was used as the loading control. The average value for sham-operated $Ncoa4^{+/+}$ hearts was set to 1 (biologically independent samples: $n = 8$). (C–H) The $Ncoa4^{+/+}$ mice were subjected to TAC and analyzed 3 days after sham operation and 3, 5, and 7 days and 4 weeks after the operation. (C) Heart homogenates after TAC were subjected to western blot analysis using anti-FTH1 antibody ($n = 5$ biologically independent samples for each group). GAPDH was used as the loading control. (D) Cardiac $Fth1$ mRNA levels after TAC ($n = 5$ biologically independent samples). $Gapdh$ mRNA was used as the loading control. (E and F) Immunofluorescence analysis of LC3B (green) and FTH1 (red) in the heart after TAC ($n = 5$ biologically independent samples). (G and H) Immunofluorescence analysis of LAMP2a (green) and FTH1 (red) in the heart after TAC ($n = 5$ biologically independent samples). Scale bar, 5 µm in (E) and (G). Arrows indicate double-positive dots. The values are presented as the mean ± SEM. The data were evaluated by one-way analysis of variance (ANOVA), followed by Tukey–Kramer's post hoc test. *$p<0.05$, **$p<0.001$, ***$p<0.0001$. NS, $p>0.05$ versus sham-operated group. Exact p-values are provided in *Supplementary file 1*.

The online version of this article includes the following source data and figure supplement(s) for figure 2:

**Source data 1.** Source data for *Figure 2*.
**Figure supplement 1.** Echocardiographic parameter aftertransverse aortic constriction (TAC).
**Figure supplement 1—source data 1.** Source data for *Figure 2—figure supplement 1*.

and TAC-operated $Ncoa4^{-/-}$ hearts (**Figure 2A**). However, the mRNA level of FTH1 was also decreased in $Ncoa4^{+/+}$ hearts (**Figure 2B**). To clarify the ferritinophagic activity, we evaluated the cardiac phenotypes during an earlier time course after pressure overload when the secondary effect to cardiac remodeling was minimal. Cardiac dysfunction and LV chamber dilation were observed in wild-type $Ncoa4^{+/+}$ hearts (**Figure 2—figure supplement 1**). Both TAC-operated $Ncoa4^{+/+}$ and $Ncoa4^{-/-}$ mice showed decreased fractional shortening 3 days after surgery compared to the corresponding sham-operated group. However, there was no significant difference in fractional shortening between TAC-operated $Ncoa4^{+/+}$ and $Ncoa4^{-/-}$ mice, suggesting mild cardiac dysfunction observed in both

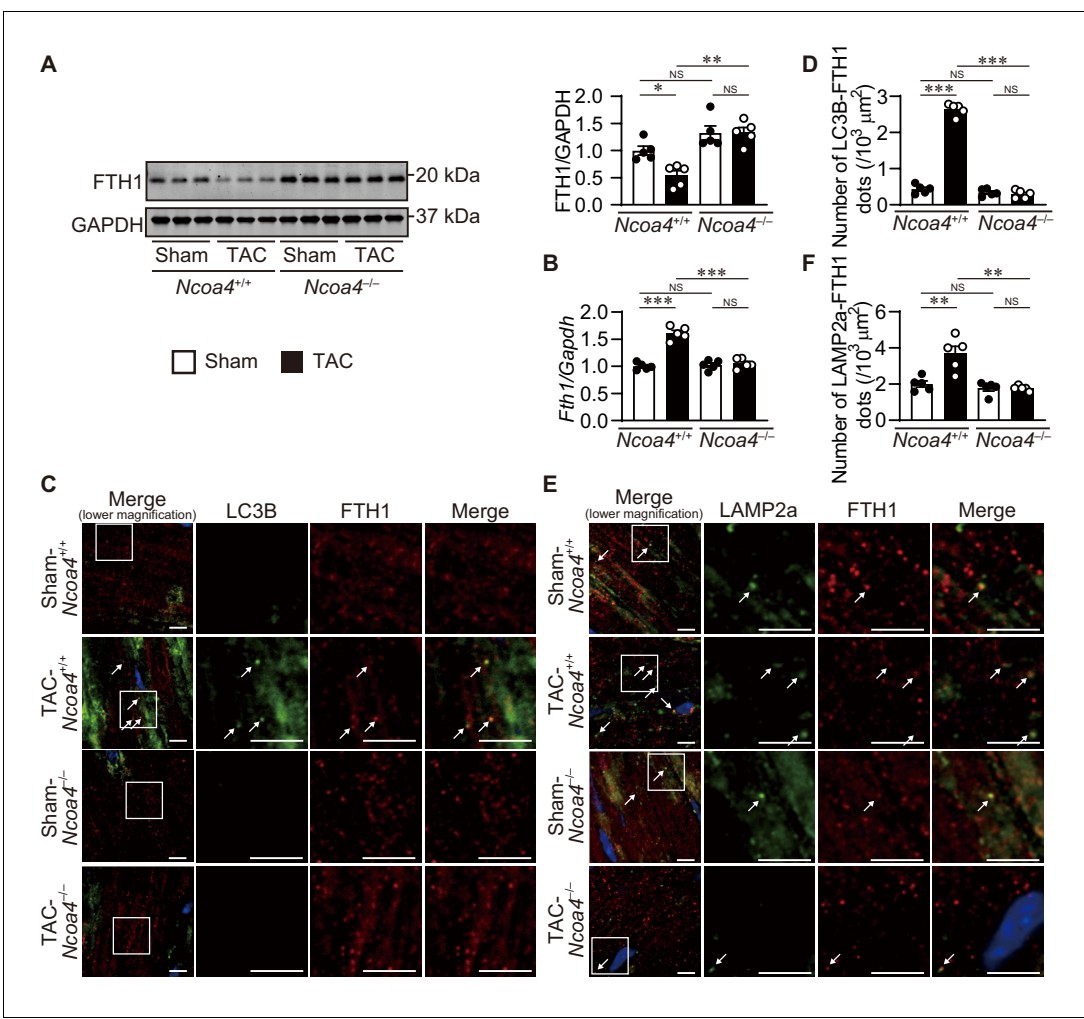

**Figure 3.** Ablation of $Ncoa4$ in cardiomyocytes showed defective ferritinophagy. (**A**) Western blot analysis of FTH1 in $Ncoa4^{+/+}$ and $Ncoa4^{-/-}$ hearts 1 week after transverse aortic constriction (TAC). GAPDH was used as the loading control. The right-hand graphs show the densitometric analysis. The average value for sham-operated $Ncoa4^{+/+}$ hearts was set to 1 (biologically independent samples: $n = 5$). (**B**) mRNA expression of $Fth1$ in the heart 1 week after TAC. $Gapdh$ mRNA was used as the loading control. The average value for sham-operated $Ncoa4^{+/+}$ hearts was set to 1 (biologically independent samples: $n = 5$). (**C and D**) Immunofluorescence analysis of LC3B (green) and FTH1 (red) in the heart 1 week after TAC. The number of LC3B- and FTH1-positive dots is shown in (**D**). (**E and F**) Immunofluorescence analysis of LAMP2a (green) and FTH1 (red) in the heart 1 week after TAC. The number of LAPM2a- and FTH1-positive dots is shown in (**F**). In (**C**) and (**E**), images of the square in the merged images are shown at higher magnification in the right three columns. Scale bar, 5 μm. Arrows indicate double-positive dots. The data were evaluated by one-way analysis of variance (ANOVA), followed by Tukey–Kramer's post hoc test. $^*p<0.05$, $^{**}p<0.001$, $^{***}p<0.0001$. NS, $p>0.05$. Exact p-values are provided in **Supplementary file 1**. The online version of this article includes the following source data for figure 3:

**Source data 1.** Source data for **Figure 3**.

groups 3 days after TAC was due to acute response to pressure overload. The protein level of FTH1 decreased from postoperative day 5 compared to sham-operated hearts, while the level of *Fth1* mRNA increased 5 and 7 days after TAC and decreased 4 weeks after TAC (*Figure 2C and D*). NCOA4 is responsible for the selective targeting of the ferritin complex to autophagosomes. In the TAC-operated hearts, the number of LC3B (a marker of an autophagosome)- and FTH1-positive dots increased after postoperative day 5 and then declined 4 weeks after TAC (day 7 versus 4 weeks, p<0.0001), and that of LAMP2a (a marker of a lysosome)- and FTH1-positive dots increased 7 days after TAC (*Figure 2E,F,G and H*).

The extent of ferritinophagy in *Ncoa4*$^{+/+}$ and *Ncoa4*$^{-/-}$ mice was then evaluated 7 days after TAC. The ablation of *Ncoa4* attenuated the downregulation of FTH1 protein and upregulation of *Fth1* mRNA in TAC-operated *Ncoa4*$^{+/+}$ hearts (*Figure 3A and B*). The number of LC3B- and FTH1-positive dots and the number of LAMP2a- and FTH1-positive dots decreased in TAC-operated *Ncoa4*$^{-/-}$ hearts compared to *Ncoa4*$^{+/+}$ hearts (*Figure 3C,D,E and F*).

## Erythropoiesis in NCOA4-deficient mice

To determine whether the failure of cardiomyocyte-specific NCOA4-deficient mice to degrade ferritin in cardiomyocytes affects erythropoiesis, as reported in global NCOA4-deficient mice (*Bellelli et al., 2016*), we explored the effect of NCOA4 deficiency in the heart on red blood cell parameters (*Figure 4—source data 2*). There were no significant differences in hematological parameters, serum iron, and transferrin saturation between any groups. Pressure overload reduced serum ferritin compared to the corresponding sham-operated group, but there was no significant difference between TAC-operated *Ncoa4*$^{+/+}$ and *Ncoa4*$^{-/-}$ mice. Serum ferritin is both a marker of liver iron stores and acute inflammatory response (*Kell and Pretorius, 2014*). TAC decreased the level of total non-heme iron in both *Ncoa4*$^{+/+}$ and *Ncoa4*$^{-/-}$ livers, but there was no significant difference in the level between *Ncoa4*$^{+/+}$ and *Ncoa4*$^{-/-}$ mice (*Figure 4—figure supplement 1A*). TAC increased the serum IL-6 level in both *Ncoa4*$^{+/+}$ and *Ncoa4*$^{-/-}$ mice, which was higher in *Ncoa4*$^{+/+}$ than that in *Ncoa4*$^{-/-}$ (*Figure 4—figure supplement 1B*). It has been reported that the serum level of IL-6 in the patients with heart failure is related to its severity (*Hirota et al., 2004*). Thus, the reduced level of serum ferritin level in TAC-operated mice may be due to the reduction in liver iron stores.

## Iron metabolism in pressure-overloaded NCOA4-deficient hearts

Next, we assessed the effect of pressure overload on iron metabolism in the heart (*Figure 4A*). Increased free ferrous iron plays a critical role in the Fenton reaction during iron-dependent necrosis. Pressure overload decreased the total non-heme iron content in both *Ncoa4*$^{+/+}$ and *Ncoa4*$^{-/-}$ hearts. The level of ferrous iron in *Ncoa4*$^{+/+}$ hearts was higher than that in *Ncoa4*$^{-/-}$ hearts under pressure overload, whereas the level of ferric iron was lower in *Ncoa4*$^{+/+}$ hearts than in *Ncoa4*$^{-/-}$ hearts. The ratio of ferrous iron to FTH1, which represents the non-binding fraction of ferrous iron to FTH1, was higher in TAC-operated *Ncoa4*$^{+/+}$ hearts than in the sham-operated controls and TAC-operated *Ncoa4*$^{-/-}$ hearts, suggesting free ferrous iron overload in TAC-operated *Ncoa4*$^{+/+}$ hearts. When iron is limited, regulatory proteins (IRPs) bind to iron regulatory elements (IREs) found in untranslated regions (UTR) of mRNA involved in iron transport and storage (*Anderson et al., 2012*). IRP binding to IREs found in the 5' UTR of mRNA encoding FTH1, FTL and ferroportin 1 (exports iron out of the cell, also known as solute carrier family 40 member 1; SLC40A1) blocks the initiation of translation. IREs, found in the 3' UTR of transferrin receptor 1 (TFRC; the membrane receptor for iron), divalent metal iron transport (solute carrier family 11 member 2; SLC11A2), cell division cycle 14A (CDC14A), and CDC binding protein kinase alpha (CDC42BPA), bind IRPs to stabilize the mRNA by inhibiting nuclease digestion. The levels of proteins related to intracellular iron metabolism such as IREB2, SLC40A1, and TFRC showed no differences between TAC-operated groups (*Figure 4—figure supplement 2A*). There were no significant differences in the mRNA levels of *Tfrc*, *Slc11a2*, *Cdc14a*, and *Cdc42bpa* between TAC-operated groups (*Figure 4—figure supplement 2B*). The binding of IRP with the 5' UTR of *Slc40a1* exhibited no difference between TAC-operated groups (*Figure 4—figure supplement 2C*). These suggest that IRP system is impaired in TAC-operated *Ncoa4*$^{+/+}$ mice.

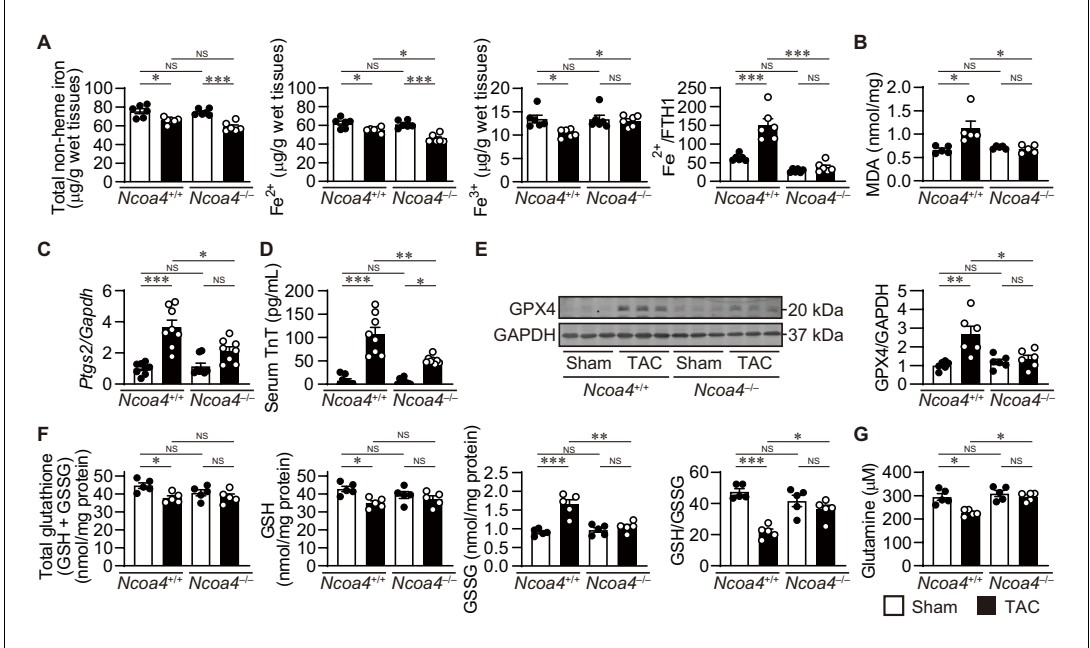

**Figure 4.** The effect of *Ncoa4* ablation on the pathways to iron-dependent cell death. (A) Tissue levels of total non-heme, ferrous, and ferric iron in *Ncoa4*[+/+] and *Ncoa4*[−/−] hearts 4 weeks after transverse aortic constriction (TAC) (*n* = 6 biologically independent samples). The ratio of the ferrous iron level to the FTH1 protein level is shown in the right-most panel. The FTH1 protein level for sham-operated *Ncoa4*[+/+] estimated as Western blot analysis in hearts was set to 1. (B) Malondialdehyde (MDA) levels in heart homogenates (*n* = 5 biologically independent samples). (C) *Ptgs2* mRNA levels in the heart (*n* = 8 biologically independent samples). (D) Serum troponin T (TnT) levels (*n* = 8 biologically independent samples). (E) Glutathione peroxidase 4 (GPX4) levels in the heart homogenates. The right panel shows the ratio of GPX4 to GAPDH (*n* = 6 biologically independent samples). (F) The levels of total glutathione (GSH+GSSG), reduced (GSH) and oxidized (GSSG) glutathione and the ratio of GSH to GSSG in heart homogenates (*n* = 5 biologically independent samples). GSH levels were calculated by subtracting GSSG from total glutathione. (G) Cardiac glutamine levels (*n* = 5 biologically independent samples). The data were evaluated by one-way analysis of variance (ANOVA), followed by Tukey–Kramer's post hoc test. *p<0.05, **p<0.001, ***p<0.0001. NS, p>0.05. Exact p-values are provided in *Supplementary file 1*.

The online version of this article includes the following source data and figure supplement(s) for figure 4:

**Source data 1.** Source data for *Figure 4*.
**Source data 2.** Hematological parameters and red cell indices in *Ncoa4*[+/+] and *Ncoa4*[−/−] mice.
**Source data 3.** Source data for data table provided in *Figure 4—source data 2*.
**Figure supplement 1.** Liver iron store and serum inflammatory cytokine in *Ncoa4*[+/+] and *Ncoa4*[−/−] mice.
**Figure supplement 1—source data 1.** Source data for *Figure 4—figure supplement 1*.
**Figure supplement 2.** Iron regulating proteins in transverse aortic constriction (TAC)-operated *Ncoa4*[−/−] hearts.
**Figure supplement 2—source data 1.** Source data for *Figure 4—figure supplement 2*.
**Figure supplement 3.** Lipid reactive oxygen species and anti-oxidant proteins in *Ncoa4*[−/−] hearts 4 weeks after transverse aortic constriction (TAC).
**Figure supplement 3—source data 1.** Source data for *Figure 4—figure supplement 3*.
**Figure supplement 4.** The system Xc−/glutathione axis and glutaminolysis pathway in transverse aortic constriction (TAC)-operated *Ncoa4*[−/−] hearts.
**Figure supplement 4—source data 1.** Source data for *Figure 4—figure supplement 4*.

## Lipid oxidation in pressure-overloaded NCOA4-deficient hearts

Lipid peroxidation is a hallmark of iron-dependent necrosis (*Dixon et al., 2012*). In *Ncoa4*[+/+] hearts, pressure overload increased the levels of malondialdehyde (MDA) and 4-hydroxy-2-nonenal (4-HNE)-positive area; markers for lipid peroxidation (*Ayala et al., 2014*; *Figure 4B* and *Figure 4—figure supplement 3A*). In contrast, these markers were attenuated in *Ncoa4*[−/−] hearts. *Ptgs2* mRNA, a putative marker for ferroptosis (*Yang et al., 2014*), was increased in TAC-operated *Ncoa4*[+/+] hearts but not in *Ncoa4*[−/−] hearts (*Figure 4C*). There were no significant differences in the antioxidant protein levels of superoxide dismutase 2 and heme oxygenase 1 between any groups (*Figure 4—figure supplement 3B*). The increase in serum troponin T (TnT), a marker for necrotic cell death, in TAC-operated *Ncoa4*[+/+] mice was significantly attenuated by *Ncoa4* ablation (*Figure 4D*). Taken

together, these results show that iron-dependent necrosis plays an important role in the development of pressure overload-induced heart failure.

## Glutathione and glutamine metabolism in pressure-overloaded NCOA4-deficient hearts

The GPX4 protein level was upregulated in TAC-operated $Ncoa4^{+/+}$ hearts compared to the corresponding controls, while ablation of $Ncoa4$ suppressed the pressure overload-induced induction of GPX4 (*Figure 4E*). The levels of total and reduced glutathione (GSH) decreased and oxidized glutathione (GSSG) increased in TAC-operated $Ncoa4^{+/+}$ hearts, resulting in a decreased GSH to GSSG ratio in $Ncoa4^{/}$ hearts (*Figure 4F*). $Ncoa4$ ablation normalized the ratio to the sham control level. The cysteine-glutamate antiporter (system Xc⁻, also known as solute carrier family 7 member 11; SLC7A11) is a key regulator for cystine uptake in cell survival against ferroptosis (*Gao et al., 2015*). There was no significant difference in the level of $Slc7a11$ mRNA or cardiac cystine between TAC-operated $Ncoa4^{+/+}$ and $Ncoa4^{-/-}$ mice (*Figure 4—figure supplement 4A and B*). The level of glutamate in TAC-operated $Ncoa4^{+/+}$ hearts was lower than that in sham-operated $Ncoa4^{+/+}$ or TAC-operated $Ncoa4^{-/-}$ mice (*Figure 4—figure supplement 4C*). L-glutamine uptake is mainly dependent on the receptors SLC38A1, SLC1A5, and SLC7A5 (*McGivan and Bungard, 2007*), and L-glutamine is converted into glutamate by glutaminase (GLS1 and GLS2). The cardiac glutamine level decreased in TAC-operated $Ncoa4^{+/+}$ hearts compared to the corresponding sham-operated mice and was lower than that in TAC-operated $Ncoa4^{-/-}$ hearts (*Figure 4G*). There were no significant differences in the mRNA levels of the glutamine transporters or glutaminases between TAC-operated groups (*Figure 4—figure supplement 4D*). The changes in glutathione metabolism and glutaminolysis in TAC-operated $Ncoa4^{+/+}$ hearts were not as seen in typical ferroptosis.

## Isoproterenol-induced cell death in isolated adult cardiomyocytes

The lipid ROS and labile iron pool during iron-dependent necrosis were further estimated using adult cardiomyocytes isolated from $Ncoa4^{+/+}$ and $Ncoa4^{-/-}$ hearts. The activation of neurohumoral factors such as catecholamine plays an important role in the pathogenesis of heart failure (*Shah and Mann, 2011*). The synthetic small-molecule compound erastin inhibits the activity of cysteine–glutamate antiporter, leading to the depletion of GSH (*Dixon et al., 2012*). High-throughput screening has identified ferrostatin-1 as a potent inhibitor of the accumulation of lipid ROS (*Dixon et al., 2012*; *Friedmann Angeli et al., 2014*; *Skouta et al., 2014*). Erastin or isoproterenol induced cell death in $Ncoa4^{+/+}$ cardiomyocytes, while this occurred to a lesser extent in $Ncoa4^{-/-}$ cardiomyocytes (*Figure 5A* and *Figure 5—figure supplement 1A*). Ferrostatin-1 inhibited both erastin- and isoproterenol-induced cardiomyocyte cell death. Treatment of $Ncoa4^{+/+}$ cardiomyocytes with either erastin or isoproterenol resulted in an increase in the cellular and lipid ROS levels, as estimated using the fluorescent probes H2DCFDA and C11-BODIPY, respectively (*Dixon et al., 2012*; *Figure 5B and C* and *Figure 5—figure supplement 1B and C*). The application of either ferrostatin-1 or $Ncoa4$ ablation prevented the generation of erastin- or isoproterenol-induced cellular and lipid ROS, which is in agreement with a previous report that used HT-1080 cells (*Dixon et al., 2012*). The labile iron pool level was measured using calcein-acetoxymethyl ester (*Yoshida et al., 2019*). Erastin and isoproterenol could both increase the level of the labile iron pool in $Ncoa4^{+/+}$ cardiomyocytes, which was attenuated by treatment with ferrostatin-1 (*Miotto et al., 2020*; *Figure 5D* and *Figure 5—figure supplement 1D*). $Ncoa4$ ablation was effective in reducing the erastin- or isoproterenol-induced upregulation of the labile iron pool. Isoproterenol decreased the protein level of FTH1 in an NCOA4-dependent manner (*Figure 5E*). A small molecule, RSL3, is another ferroptosis inducer, which binds and inhibits GPX4 (*Yang et al., 2014*). RSL3 induced cardiomyocyte death, which was attenuated by $Ncoa4$ ablation or ferrostatin-1 treatment (*Figure 5—figure supplement 2*).

## Attenuation of the development of cardiac remodeling by ferrostatin-1

To examine the involvement of iron-dependent necrosis in the pathogenesis of heart failure and whether iron-dependent necrosis is a therapeutic target for the disease, wild-type C57BL/6J mice received an intraperitoneal daily injection of ferrostatin-1. Four weeks after TAC, saline-administered mice exhibited LV chamber dilation and cardiac dysfunction (*Figure 6A and B*). Ferrostatin-1 administration significantly reduced the LV chamber size and improved cardiac function in TAC-operated

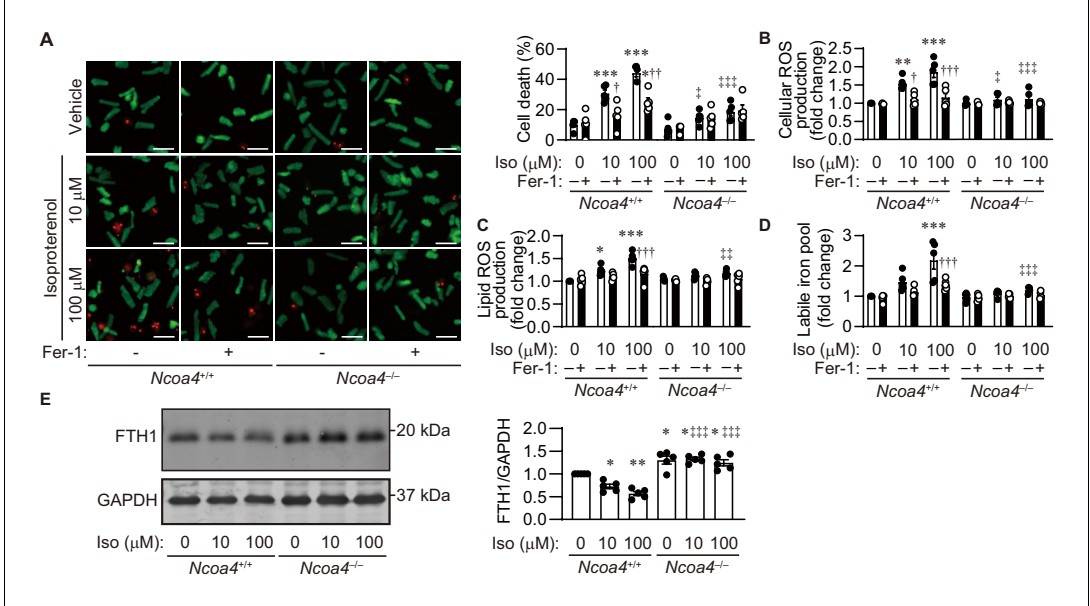

**Figure 5.** Isoproterenol induces iron-dependent cell death in isolated cardiomyocytes. (A) Cell death assay. Cell death was estimated using a Live/Dead Viability/Cytotoxicity Assay Kit. Isolated mouse cardiomyocytes from $Ncoa4^{+/+}$ and $Ncoa4^{-/-}$ hearts were treated with the indicated concentrations of isoproterenol (Iso) with or without ferrostatin-1 (Fer-1) for 4 hr. Calcein-AM (green) is retained in live cells, while ethidium homodimer produces red fluorescence in dead cells. Scale bar, 100 µm. The percentage of dead cells is shown in the middle left-hand graphs (n = 5 biologically independent samples). (B and C) The accumulation of cellular (B) and lipid (C) reactive oxygen species (ROS) were assessed by H2DCFDA and C11-BODIPY, respectively (n = 5 biologically independent samples). (D) The labile iron pool was measured using the calcein-AM method in isolated mouse cardiomyocytes (n = 5 biologically independent samples). (E) Western blot analysis of FTH1 in isolated mouse cardiomyocytes from $Ncoa4^{+/+}$ and $Ncoa4^{-/-}$ hearts. GAPDH was used as the loading control. The right-hand graphs show the densitometric analysis (n = 5 biologically independent samples). The average value for vehicle-treated without ferrostatin-1 $Ncoa4^{+/+}$ cardiomyocytes was set to 1. The values are presented as the mean ± SEM. Two-way analysis of variance (ANOVA) followed by Tukey's multiple comparisons test was used. $^*p<0.05$, $^{**}p<0.001$, $^{***}p<0.0001$. NS, $p>0.05$ versus $Ncoa4^{+/+}$ control without ferrostatin-1 treatment. $^†p<0.05$, $^{††}p<0.001$, $^{†††}p<0.0001$. NS, $p>0.05$ versus the corresponding group without ferrostatin-1 treatment. $^‡p<0.05$, $^{‡‡}p<0.001$, $^{‡‡‡}p<0.0001$. NS, $p>0.05$ versus the corresponding $Ncoa4^{+/+}$. Exact p-values are provided in *Supplementary file 1*.

The online version of this article includes the following source data and figure supplement(s) for figure 5:

**Source data 1.** Source data for *Figure 5*.
**Figure supplement 1.** Erasin induces cell death in isolated cardiomyocytes.
**Figure supplement 1—source data 1.** Source data for *Figure 5—figure supplement 1*.
**Figure supplement 2.** RSL3 induces cell death in isolated cardiomyocytes.
**Figure supplement 2—source data 1.** Source data for *Figure 5—figure supplement 2*.

mice. Pressure overload-induced increases in LV mass and weight, the cross-sectional area of cardiomyocytes, and remodeling markers such as *Nppa*, *Nppb*, and *Myh7* mRNAs were significantly attenuated in ferrostatin-1-treated hearts (*Figure 6C and D* and *Figure 6—figure supplement 1A*). TAC-operated saline-treated mice exhibited cardiac fibrosis, which was diminished by ferrostatin-1 (*Figure 6D* and *Figure 6—figure supplement 1A*). Thus, ferrostatin-1 prevented the development of pressure overload-induced cardiomyopathy. TAC-operated control mice showed increased lipid ROS and *Ptgs2* mRNA, which was inhibited by the administration of ferrostatin-1 (*Figure 6E and F* and *Figure 6—figure supplement 1B*). Taken together, these findings indicate that iron-dependent necrosis was involved in the pathogenesis of pressure overload-induced heart failure in the wild-type mice.

To examine whether iron-dependent cell death is downstream of NCOA4, $Ncoa4^{-/-}$ mice received an intraperitoneal daily injection of ferrostatin-1. Four weeks after TAC, there was no significant difference in the extent of cardiac remodeling between saline- and ferrostatin-1-treated mice (*Figure 6—figure supplement 2*).

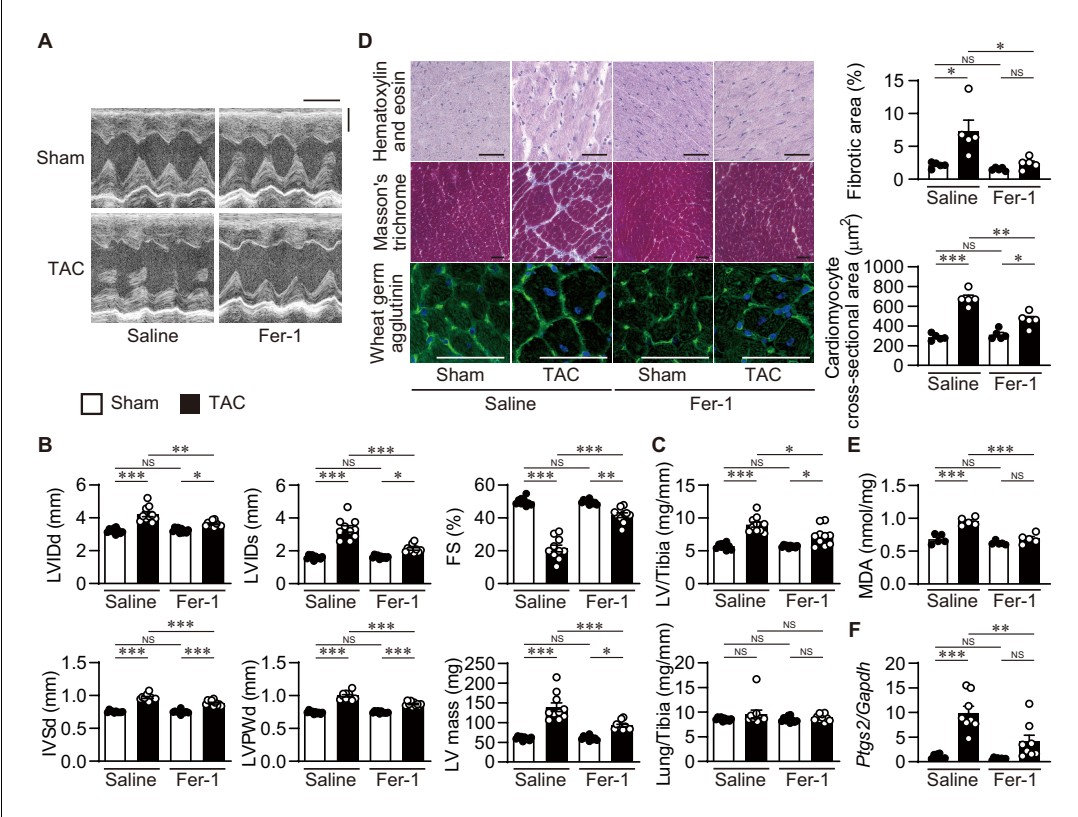

**Figure 6.** Inhibition of lipid peroxidation protects hearts from pressure overload. Wild-type C57BL/6J mice were subjected to transverse aortic constriction (TAC) and analyzed 4 weeks after the operation. Ferrostatin-1 (Fer-1) or saline was intraperitoneally administered daily starting 1 day before TAC. (A) Representative images of transthoracic M-mode echocardiographic tracing. Scale bars, 0.1 s and 2 mm, respectively. (B) Echocardiographic parameters of the mice (n = 10 biologically independent samples). (C) Physiological parameters of the mice (n = 10 biologically independent samples). (D) Histological analysis of the heart. Scale bar, 50 μm. The upper and lower right graphs show the ratio of the fibrotic area to whole heart section and the cross-sectional area of cardiomyocytes, respectively (n = 5 biologically independent samples). (E) Cardiac MDA levels (n = 5 biologically independent samples). (F) *Ptgs2* mRNA levels in the hearts (n = 8 biologically independent samples). The data were evaluated by one-way analysis of variance (ANOVA) followed by Tukey–Kramer's post hoc test. *p<0.05, **p<0.001, ***p<0.0001. NS, p>0.05. Exact p-values are provided in ***Supplementary file 1***.

The online version of this article includes the following source data and figure supplement(s) for figure 6:

**Source data 1.** Source data for *Figure 6*.

**Figure supplement 1.** Cardiac remodeling markers and 4-HNE staining in ferrostatin-1-treated transverse aortic constriction (TAC)-operated wild-type hearts.

**Figure supplement 1—source data 1.** Source data for *Figure 6—figure supplement 1*.

**Figure supplement 2.** Ferrostatin-1 does not provide additional protection from cardiac remodeling in *Ncoa4*⁻/⁻ mice.

**Figure supplement 2—source data 1.** Source data for *Figure 6—figure supplement 2*.

## Discussion

Our data indicate that there is no cardiomyocyte-autonomous requirement for NCOA4 during normal embryonic development. Furthermore, the NCOA4-mediated pathway does not appear to be required for normal heart growth in the postnatal period. We found that NCOA4-dependent ferritinophagy is activated for degradation of ferritin in the heart in response to pressure overload and is detrimental in the stressed heart. The release of ferrous iron from ferritin leads to the increase in lipid ROS, cardiac necrosis, and heart failure.

We observed the reduction of FTH1 protein level in *Ncoa4*⁺/⁺ hearts 4 weeks after TAC as we previously reported (*Omiya et al., 2009*). Because the mRNA level of FTH1 was also decreased, it was not conclusive that the decrease in FTH1 was due to increased ferritinophagy. Our study analyzing the heart during an earlier time course after pressure overload showed that the protein level of

FTH1 decreased along with an increase in the level of *Fth1* mRNA, suggesting that the downregulation of FTH1 is due to its degradation. In the TAC-operated hearts, FTH1 was recruited in autophagosomes or autolysosomes and this recruitment was NCOA4-dependent. These findings confirmed that pressure overload induces the activation of NCOA4-mediated ferritinophagy, which is detrimental to the heart. The labile iron pool and oxidative stress are known to increase *Fth1* mRNA (*Arosio et al., 2009*), which may explain the increased level of *Fth1* mRNA at the early time points after pressure overload. The detrimental effect of FTH1 downregulation on cardiac function is supported by a recent report showing that mice lacking FTH1 in cardiomyocytes increased oxidative stress, resulting in mild cardiac dysfunction upon aging (*Fang et al., 2020*).

Iron is essential for the survival of the cells, as it serves as a cofactor in the biochemical processes such as oxygen storage, oxidative phosphorylation, and enzymatic reaction (*Andrews and Schmidt, 2007*). Intracellular iron levels are maintained by NCOA4-dependent ferritin degradation (*Mancias et al., 2014*). While ferritin supplies iron for mitochondrial function (*Fujimaki et al., 2019*), it protects the cell against free radical generation via the Fenton reaction (*Papanikolaou and Pantopoulos, 2005*). These suggest the importance of NCOA4 in intracellular iron homeostasis and a double-edged sword role of ferritinophagy. Our study presented here indicate that NCOA4-mediated ferritinophagy is pathologic for the heart in response to pressure overload by activating iron-dependent cell death. In addition, *Ncoa4*$^{-/-}$ mice showed normal cardiac function at baseline as well as under hemodynamic stress, suggesting that iron derived from ferritin degradation is not necessary for cardiac homeostasis or some other transit pools of iron such as low-molecular-weight chelates including citrate, ATP, AMP, or pyrophosphate may compensate the loss of NCOA4 in the heart (*Papanikolaou and Pantopoulos, 2005*). Why does the maladaptive hyperactivation of ferritinophagy occur in pressure overloaded hearts? NCOA4-dependent ferritinophagy is regulated by intracellular iron (*Dowdle et al., 2014*; *Mancias et al., 2014*). However, the precise activation mechanism of ferritinophagy–cell death pathway in response to hemodynamic stress remains to be elucidated. We found that β$_1$-adrenergic agonist, isoproterenol, induced cardiomyocyte death in which the NCOA4-dependent pathway, the generation of lipid ROS, and increased labile iron pool are involved. In addition, we showed that isoproterenol induces ferritinophagy. Thus, the downstream signaling pathways of β$_1$-adrenergic receptors activate ferritinophagy and resultant cell death. Another possibility is that isoproterenol induces intracellular iron deficiency in cardiomyocytes to lead to ferritinophagy. However, the expression level of FTH1 was maintained in cardiomyocyte-specific IREB1/2-deficient hearts with iron deficiency (*Haddad et al., 2016*), suggesting that iron deficiency may not directly induce FTH1 degradation in cardiomyocytes. It is also possible that NCOA4 may associate with NRF2/HO-1 and VDACs-induced mitochondrial dysfunction pathways in ferroptosis-induced cardiomyocytes death and heart failure (*Li et al., 2020*). It has been reported that NCOA4 has association with mitochondria iron-overload in cardiomyocyte hypertrophy pathophysiology (*Tang et al., 2019*). Although NCOA4-mediated ferritin degradation contributes to maintain mitochondrial function through iron supply (*Fujimaki et al., 2019*), excessive ferritinophagy may induce cardiomyocyte hypertrophy and cell death. Further studies are necessary to elucidate molecular mechanism underlying NCOA4-mediated cardiomyocyte death and heart failure.

Although pressure overload decreased the level of ferrous iron in the heart, the level of ferrous iron in *Ncoa4*$^{+/+}$ hearts was higher than that in *Ncoa4*$^{-/-}$ hearts. The calculated non-binding fraction of ferrous iron to FTH1 was higher in TAC-operated *Ncoa4*$^{+/+}$ hearts. To confirm the increased level of ferrous iron in pressure overloaded *Ncoa4*$^{+/+}$ hearts, we measured the labile iron pool in isolated cardiomyocytes. Erastin and isoproterenol could both increase the level of the labile iron pool in *Ncoa4*$^{+/+}$ cardiomyocytes, which was attenuated by *Ncoa4* ablation. These suggest labile iron overload induced by activation of ferritinophagy in stressed cardiomyocytes, even though the pressure-overloaded *Ncoa4*$^{+/+}$ mice exhibited decreased total cardiac iron.

Deregulation of programmed cardiomyocyte death has been reported to play an important role in the pathogenesis of heart failure (*Whelan et al., 2010*). While apoptosis is the best-studied form of programmed cell death, there are also non-apoptotic programmed cell death. Necroptosis and iron-dependent necrosis are two distinct regulated necrotic cell death. We observed increases in lipid peroxidation in TAC-operated *Ncoa4*$^{+/+}$ hearts but not in *Ncoa4*$^{-/-}$ hearts. In addition, ferrostatin-1 attenuated the development of pressure overload-induced cardiac remodeling. Taken together, these results show that iron-dependent necrosis plays an important role in the development of pressure overload-induced heart failure. Ferrostatin-1 did not provide additional protection

from pressure overload-induced cardiac remodeling in $Ncoa4^{-/-}$ mice, suggesting that iron-dependent cardiomyocyte death is downstream of NCOA4-mediated ferritinophagy. GPX4 inhibits the formation of lipid peroxidation and ferroptosis (*Yang et al., 2014*), while glutaminolyis is required for the execution of ferroptosis. We found that *Ptgs2* mRNA, a putative marker for ferroptosis, was upregulated TAC-operated $Ncoa4^{+/+}$ hearts; however, the GPX4 protein level was upregulated, while glutamate and glutamine were downregulated in TAC-operated $Ncoa4^{+/+}$ hearts. The mRNA of *Slc7a11* in system Xc⁻ and cystine showed no difference between TAC-operated $Ncoa4^{+/+}$ and $Ncoa4^{-/-}$ hearts. These results suggest that the pressure overload-induced increase in GPX4 was compensatory to prevent iron-dependent necrosis, and insufficient induction of GPX4 may lead to the increase in lipid ROS and the downregulation of glutamine or glutamate was also a compensatory mechanism to inhibit iron-dependent necrosis. ROS have a variety of physiological and pathological functions depending on their source, species and local concentration, local antioxidant environment, and possibly the disease stage (*Papaharalambus and Griendling, 2007*). Intensive studies have implicated ROS in the development of cardiovascular pathology including cardiac remodeling (*Ponikowski et al., 2016*). However, the failure of clinical trials using antioxidants requests more precise understanding of the sources and contribution of ROS in heart failure. Our study indicates that lipid ROS derived from ferritinophagy and the Fenton reaction plays an important role in the pathogenesis of heart failure. Thus, our study supports the notion that inhibiting lipid peroxidation is cardioprotective during pressure overload.

Iron deficiency is a common condition affecting approximately 50% of patients with heart failure (*Lavoie, 2020*). Clinical trials have demonstrated the symptomatic benefit of treating iron-deficient heart failure patients with intravenous iron supplementation (*Anker et al., 2009*; *Ghafourian et al., 2020*). However, heart failure patients may have underlying myocardial iron overload (*Ghafourian et al., 2020*; *Sawicki et al., 2015*). In addition, a high-iron diet caused severe cardiac dysfunction in cardiomyocyte-specific FTH1-deficient mice (*Fang et al., 2020*). These raises concerns about the safety of the prolonged use of iron supplementation in heart failure patients. The long-term safety of iron supplementation in heart failure patients must be determined. Our results do not contradict the clinical trials but rather imply a potential role for reducing iron-dependent cell death in patients with heart failure.

In summary, the data presented here reveal a novel mechanism underlying the pathogenesis of heart failure. Iron-dependent cardiomyocyte death plays an important role in the development of pressure overload-induced heart failure. The inhibition of iron-dependent cardiomyocyte death can be a novel therapeutic mechanism for patients with heart failure.

## Materials and methods

**Key resources table**

| Reagent type (species) or resource | Designation | Source or reference | Identifiers | Additional information |
|---|---|---|---|---|
| Strain, strain background (male *Mus musculus*) | C57BL/6J | Envigo UK | | C57BL/6OlaHsd |
| Strain, strain background (male *Mus musculus*) | $Ncoa4^{flox/flox}$;Myh6-Cre⁺ | This paper | | See Materials and methods |
| Strain, strain background (male *Mus musculus*) | $Ncoa4^{flox/flox}$;Myh6-Cre⁻ | This paper | | See Materials and methods |
| Strain, strain background (male *Mus musculus*) | Myh6-Cre⁺ | *Nishida et al., 2004* | | See Materials and methods |
| Antibody | Mouse monoclonal antibody to NCOA4 | Sigma-Aldrich | SAB1404569, Lot: HC071-1F11, RRID:AB_10759525 | (1:1000) |

*Continued on next page*

Continued

| Reagent type (species) or resource | Designation | Source or reference | Identifiers | Additional information |
|---|---|---|---|---|
| Antibody | Rabbit polyclonal antibody to FTH1 | Cell Signaling Technology | 3998, Lot: 2, RRID:AB_1903974 | for western blots, (1:1000); for immunofluorescence, (1:100) |
| Antibody | Mouse monoclonal antibody to LC3B | Cell Signaling Technology | 83506, Lot: 1, RRID:AB_2800018 | (1:100) |
| Antibody | Rat monoclonal antibody to LAMP2a | Abcam | ab13524, Lot: GR3317907-1, RRID:AB_2134736 | (1:100) |
| Antibody | Rat monoclonal antibody to GPX4 | Millipore | MABS1274, Lot: Q2633070, RRID:AB_2885039 | (1:1000) |
| Antibody | Rabbit polyclonal antibody to 4-HNE | Millipore | 393207, Lot: 3167038, RRID:AB_566310 | (1:500) |
| Antibody | Rabbit polyclonal antibody to IREB2 | Thermo Fisher Scientific | PA1-16543, Lot: TK2666362A, RRID:AB_2126583 | (1:250) |
| Antibody | Rabbit polyclonal antibody to SLC40A1 | Alpha Diagnostic International | MTP11-A, Lot: 1169899A3-L, RRID:AB_1619475 | (1:1000) |
| Antibody | Mouse monoclonal antibody to TFRC | Thermo Fisher Scientific | 13–6800, Lot: TI275369, RRID:AB_2533029 | (1:1000) |
| Antibody | Rabbit polyclonal antibody to SOD2 | Abcam | ab13534, Lot: GR33618-66, RRID:AB_2191667 | (1:1000) |
| Antibody | Rabbit polyclonal antibody to HO-1 | Enzo Life Sciences | ADI-SPA-895, Lot: 03301708, RRID:AB_10618757 | (1:1000) |
| Antibody | Mouse monoclonal antibody to GAPDH | Sigma-Aldrich | G8795, Lot: 056M4856V, RRID:AB_1078991 | (1:10000) |
| Antibody | Mouse immunoglobulin | Santa Cruz biochemistry | sc-2025, RRID:AB_737182 | |
| Antibody | Rabbit IgG | Abcam | ab37415, RRID:AB_2631996 | |
| Antibody | Fluorescein isothiocyanate-conjugated lectin | Sigma-Aldrich | L4895 | |
| Antibody | Alexa Fluor 488 donkey-anti-mouse | Thermo Fisher Scientific | A21202, RRID:AB_141607 | (1:500) |
| Antibody | Alexa Fluor 568 donkey-anti-rabbit | Thermo Fisher Scientific | A10042, RRID:AB_2534017 | (1:500) |
| Antibody | Alexa Fluor 488 goat-anti-rat | Thermo Fisher Scientific | A11006, RRID:AB_2534074 | (1:500) |
| Antibody | IRDye 680LT Donkey anti-Mouse IgG Secondary Antibody | LI-COR Biosciences | 926–68020, RRID:AB_10706161 | (1:10,000) |
| Antibody | IRDye 680LT Donkey anti-Rabbit IgG Secondary Antibody | LI-COR Biosciences | 926–68023, RRID:AB_10706167 | (1:10,000) |
| Antibody | IRDye 680LT Goat anti-Rat IgG Secondary Antibody | LI-COR Biosciences | 926–68029, RRID:AB_10715073 | (1:10,000) |
| Commercial assay or kit | Pierce Protein G Magnetic Beads | Thermo Scientific | 88848 | |
| Commercial assay or kit | RNeasy Fibrous Tissue Mini Kit | QIAGEN | 74704 | |
| Commercial assay or kit | SuperScript IV First-Strand Synthesis System | Thermo Fisher Scientific | 18091050 | |
| Commercial assay or kit | PowerUp SYBR Green Master Mix | Thermo Fisher Scientific | A25742 | |
| Commercial assay or kit | Masson's Trichrome Stain Kit | Polysciences, Inc | 25088–1 | |

*Continued*

| Reagent type (species) or resource | Designation | Source or reference | Identifiers | Additional information |
|---|---|---|---|---|
| Commercial assay or kit | VECTASTAIN Elite ABC-HRP Kit, Peroxidase (Rabbit IgG) | Vector Laboratories Inc | PK-6101 | |
| Commercial assay or kit | DAB Substrate Kit, Peroxidase (HRP), with Nickel, (3,3'-diaminobenzidine) | Vector Laboratories Inc | SK-4100 | |
| Commercial assay or kit | normal donkey serum | Abcam | ab7475, RRID:AB_2885042 | |
| Commercial assay or kit | normal goat serum | Abcam | ab7481, RRID:AB_2716553 | |
| Commercial assay or kit | ProLong Gold Antifade Reagent with DAPI | Life Technologies | P36935 | |
| Commercial assay or kit | Mouse Ferritin ELISA Kit (FTL) | Abcam | ab157713 | |
| Commercial assay or kit | Pointe Scientific Iron/TIBC Reagents | Pointe Scientific | 23-666-320 | |
| Commercial assay or kit | ELISA Kit for Troponin T Type 2, Cardiac (TNNT2) | Cloud-Clone | SED232Mu | |
| Commercial assay or kit | Mouse IL-6 Quantikine ELISA Kit | R and D Systems | M6000B | |
| Commercial assay or kit | Iron Assay Kit | Abcam | ab83366 | |
| Commercial assay or kit | Lipid Peroxidation (MDA) Assay Kit | Abcam | ab118970 | |
| Commercial assay or kit | GSSG/GSH Quantification Kit | Dojindo | G257 | |
| Commercial assay or kit | Glutamine Assay Kit | Abcam | ab197011 | |
| Commercial assay or kit | Glutamate Assay Kit | Abcam | ab83389 | |
| Commercial assay or kit | Electrophoretic Mobility-Shift Assay (EMSA) Kit | Invitrogen | E33075 | |
| Commercial assay or kit | Zero Blunt TOPO PCR Cloning Kit | Invitrogen | 451245 | |
| Commercial assay or kit | HiScribe T7 Quick High Yield RNA Synthesis Kit | New England Biolabs | E2050S | |
| Chemical compound, drug | Ferrostatin-1 | Sigma Aldrich | SML0583 | |
| Chemical compound, drug | Isoprenaline hydrochloride | Sigma Aldrich | I5627 | |
| Chemical compound, drug | Erastin | Sigma Aldrich | E7781 | |
| Chemical compound, drug | 1S,3R-RSL 3 | Sigma Aldrich | SML2234 | |
| Chemical compound, drug | calcein-AM | Invitrogen | C1430 | |
| Chemical compound, drug | ethidium homodimer-1 | Invitrogen | E1169 | |
| Chemical compound, drug | 2', 7'-dichlorodihydrofluorescein diacetate (H2DCFDA) | Invitrogen | D399 | |
| Chemical compound, drug | C11-BODIPY | Invitrogen | D3861 | |
| Chemical compound, drug | pyridoxal isonicotinoyl hydrazine (PIH) | Abcam | ab145871 | |
| Software, algorithm | ImageJ | National Institutes of Health | Version 1.51 r, RRID:SCR_003070 | |
| Software, algorithm | GraphPad Prism 8 | GraphPad Software | RRID:SCR_002798 | |
| Software, algorithm | EZChrom Elite | Agilent Technologies | Version 3.3.2. | |

## Animal studies

All procedures were carried out in accordance with the King's College London Ethical Review Process Committee and the UK Home Office (Project License No. PPL70/8889) and were performed in accordance with the Guidance on the Operation of the Animals (Scientific Procedures) Act, 1986 (UK Home Office).

## Antibodies

The following antibodies were used in this study: monoclonal mouse antibody to NCOA4 (Sigma-Aldrich, SAB1404569, Lot: HC071-1F11, 1/1000), polyclonal rabbit antibody to FTH1 (Cell Signaling Technology, 3998, Lot: 2, for western blots, 1/1000; for immunofluorescence, 1/100), monoclonal mouse antibody to LC3B (Cell Signaling Technology: 83506, Lot: 1, 1/100), monoclonal rat antibody to LAMP2a (Abcam: ab13524, Lot: GR3317907-1, 1/100), monoclonal rat antibody to GPX4 (Millipore: MABS1274, Lot: Q2633070, 1/1000), polyclonal rabbit antibody to 4-HNE (Millipore: 393207, Lot: 3167038, 1/500), polyclonal rabbit antibody to IREB2 (Thermo Fisher Scientific: PA1-16543, Lot: TK2666362A, 1/250), polyclonal rabbit antibody to SLC40A1 (Alpha Diagnostic International: MTP11-A, Lot: 1169899A3-L, 1/1000), monoclonal mouse antibody to TFRC (Thermo Fisher Scientific: 13–6800, Lot: TI275369, 1/1000), polyclonal rabbit antibody to SOD2 (Abcam: ab13534, Lot: GR33618-66, 1/1000), polyclonal rabbit antibody to HO-1 (Enzo Life Sciences: ADI-SPA-895, Lot: 03301708, 1/1000), monoclonal mouse antibody to GAPDH (Sigma-Aldrich: G8795, Lot: 056M4856V, 1/10,000).

## Generation of cardiomyocyte-specific NCOA4-deficient mice

The *Ncoa4* gene-targeting vector was constructed using mouse C57BL/6J genomic DNA (*Misaka et al., 2018*). The targeting vector was electroporated into ES cells (F1; SVJ129 and C57BL/6J), and the transfected ES clones were selected for neomycin resistance according to standard protocols. The neomycin-resistant ES clones with targeted homologous recombination were screened by PCR and further confirmed by Southern blotting. Circular pCAG-Flpe plasmid and pPGK-Puro plasmid were electroporated into the selected ES clones, and the transfected ES clones were selected for puromycin resistance according to standard protocols. The neomycin cassette-excised ES clones were screened by PCR. Southern blotting and karyotyping analyses were performed to obtain ES clones exhibiting the desired homologous recombination and normal karyotype. These targeted ES clones were injected into blastocyst C57BL/6J mouse embryos to generate chimeric mice. The chimeric mice were crossed with C57BL/6J mice to validate germ line transmission. We generated mice with the floxed *Ncoa4* allele and crossed them with transgenic mice expressing α-myosin heavy chain promoter-driven Cre recombinase (Myh6-Cre) to obtain cardiomyocyte-specific NCOA4-deficient mice (*Ncoa4*<sup>flox/flox</sup>;Myh6-Cre⁺) (*Nishida et al., 2004*). *Ncoa4*<sup>flox/flox</sup>;Myh6-Cre⁻ littermates were used as controls. The mice had access to food and water ad libitum.

## Immunoprecipitation and western blot analysis

To evaluate NCOA4 protein expression level in hearts, the protein was immunoprecipitated with an anti-NCOA4 antibody, followed by immunoblot with the antibody. One hundred micrograms protein homogenates with lysis buffer (50 mmol/L Tris-HCl, 50 mmol/L NaCl, 1 mmol/L EDTA, 1% NP-40, a protease inhibitor cocktail, pH 7.4) were precleared with 20 µL of magnetic beads-coupled protein G (Thermo Fisher Scientific, 1004D). Precleared homogenates were subjected to immunoprecipitation using 1 µg of the anti-NCOA4 antibody (Sigma-Aldrich, SAB1404569) or mouse immunoglobulin G (IgG; Santa Cruz biochemistry, sc-2025) and 40 µL of magnetic beads-coupled protein G at 4 ˚C for 2 hr. The precipitated complexes were washed three times with lysis buffer. Protein homogenates with lysis buffer were extracted from the left ventricles. The precipitated complexes or 5–15 µg of total protein homogenates were subjected to western blot analysis. After incubation with secondary antibody, the blot was developed with an infrared imaging system (ODYSSEY CLx; LI-COR Biosciences). Image Studio software (LI-COR Biosciences) was used for quantitative analysis to evaluate protein expression levels.

## Real-time quantitative reverse transcription PCR

Total RNA was isolated from the left ventricles using RNeasy Fibrous Tissue Mini Kit (QIAGEN). The mRNA expression levels were determined by quantitative reverse transcription polymerase chain reaction (PCR) using SuperScript IV reverse transcriptase (Thermo Fisher Scientific Inc) for reverse transcription and a PowerUp SYBR Green PCR Master Mix (Thermo Fisher Scientific) for the quantitative PCR reaction with the following PCR primers: forward 5'-CTATATCCAGGTGCCAGAGCAG-3' and reverse 5'-TTGCTTACAAGAAGCCACTCAC-3' for *Ncoa4*, forward 5'-TGGAGTTGTATGCCTCCTACG-3' and reverse 5'-TGGAGAAAGTATTTGGCAAAGTT-3' for *Fth1*, forward 5'-CAGACAACA

TAAACTGCGCCTT-3' and reverse 5'-GATACACCTCTCCACCAATGACC-3' for *Ptgs2*, forward 5'-TGGCCAGCAAGATTGTGGAGAT-3' and reverse 5'-TTTGCGGGTGAAGAGGAAGT-3' for *Slc1a5*, forward 5'-ATGGAGTGTGGCATTGGCTT-3' and reverse 5'-TGCATCAGCTTCTGGCAGAGCA-3' for *Slc7a5*, forward 5'-TCTACAGGATTGCGAACATCT-3' and reverse 5'-CTTTGTCTAGCATGACACCATCT-3' for *Gls1*, forward 5'-AGCGTATCCCTATCCACAAGTTCA-3' and reverse 5'-GCAGTCCAGTGGCCTTCAGAG-3' for *Gls2*, forward 5'-TCGTCTTGGCCTTTTGGCT–3' and reverse 5'-TCCAGGTGGTCTAGCAGGTTCT-3' for *Nppa*, forward 5'-AAGTCCTAGCCAGTCTCCAGA-3' and reverse 5'-GAGCTGTCTCTGGGCCATTTC-3' for *Nppb*, forward 5'-ATGTGCCGGACCTTGGAAG-3' and reverse 5'-CCTCGGGTTAGCTGAGAGATCA-3' for *Myh7*, forward 5'-ACGCGGACTCTGTTGCTGCT-3' and reverse 5'-GCGGGACCCCTTTGTCCACG-3' for *Col1a2*, forward 5'-CCCGGGTGCTCCTGGA-CAGA-3' and reverse 5'-CACCCTGAGGACCAGGCGGA-3' for *Col3a1*, forward 5'-TGCAATCTGCATCTCCATGGCT-3' and reverse 5'-AAGCAGGAGAGGGCAACAAA-3' for *Slc7a11*, forward 5'-TGGAATCCCAGCAGTTTCTT-3' and reverse 5'-GCTGCTGTACGAACCATTTG-3' for *Tfrc*, forward 5'-GGCTTTCTTATGAGCATTGCCTA-3' and reverse 5'-GGAGCACCCAGAGCAGCTTA-3' for Slc11a2, forward 5'-TGGACCTCTGAACTTGGCAAT-3' and reverse 5'-AGATGACGGCATAAGCACCTAT-3' for *Cdc14a*, forward 5'-TTTCCACCTAAGCGCAAGACT-3' and reverse 5'-ATGACATGA-GAACCCACAGA-3' for *Cdc42bpa*, and forward 5'-ATGACAACTTTGTCAAGCTCATTT-3' and reverse 5'-GGTCCACCACCCTGTTGCT-3' for *Gapdh*. PCR standard curves were constructed using the corresponding cDNA and all data were normalized to the *Gapdh* mRNA content and are expressed as the fold increase over the control group.

## Transverse aortic constriction (TAC) and echocardiography

The 8– to 12 week-old male mice were subjected to TAC using a 26-gauge needle or to a sham surgery, as previously reported (*Omiya et al., 2020*). In TAC, a small piece of 6–0 silk suture was placed between the innominate and left carotid arteries. Three loose knots were tied around the transverse aorta, and a 26-gauge needle was placed parallel to the transverse aorta. The knots were tied quickly against the needle and the needle was removed promptly to yield a 26-gauge stenosis. Sham surgeries were identical except for the aortic constriction. Echocardiography was conducted with a Vevo 2100 system (Visual Sonics) on conscious mice (*Omiya et al., 2020*). Noninvasive measurement of the tail blood pressure was also performed on conscious mice using a NP-NIBP Monitor for mice and rats (Muromachi Kikai), as previously described (*Omiya et al., 2020*).

## Histological analysis

Left ventricle samples were embedded in OCT compound (Thermo Fisher Scientific Inc) and then immediately frozen in liquid nitrogen. The samples were sectioned into 5 μm thick sections. The sections were fixed with acetone for hematoxylin–eosin staining and Masson's trichrome staining, with 4% paraformaldehyde for wheat germ agglutinin staining and with Bouin's solution for 4-HNE staining. Hematoxylin–eosin staining and Masson's trichrome staining (Masson's Trichrome Stain Kit, Polysciences Inc) were performed on serial sections. For wheat germ agglutinin staining, heart samples were stained with fluorescein isothiocyanate-conjugated lectin (Sigma, L4895) to measure the cross-sectional area of cardiomyocytes. For 4-HNE staining, rabbit anti-4-HNE antibody or control rabbit IgG (Abcam, ab37415) were used as primary antibody, and avidin-peroxidase (Vectastain Elite ABC Kit; Vector Laboratories Inc) and the DAB Peroxidase Substrate Kit (Vector Laboratories Inc) were applied, followed by counterstaining with hematoxylin as described previously (*Omiya et al., 2020*). Images were captured by an All-in-one fluorescence microscope (BZ-X700, Keyence). Quantitative analyses of the fibrosis fraction and 4-HNE positive area were examined in whole left ventricles and cardiomyocyte cross-sectional areas we examined in five different areas per section and measured using ImageJ (National Institutes of Health; Version 1.51 r).

## Immunofluorescence microscopy

The OCT-compound embedded frozen left ventricle samples were used to detect LC3B-FTH1 and LAMP2a-FTH1 co-localization dots. The samples were sectioned into 5 μm thick sections and fixed with 4% paraformaldehyde for immunohistochemical fluorescence staining. The samples were blocked with 10% normal donkey serum (Abcam, ab7475) to detect LC3B-FTH1 co-localization and with 10% normal donkey serum and 10% normal goat serum (Abcam, ab7481) to detect LAMP2a-

FTH1 co-localization. The primary antibodies were rabbit anti-FTH1, mouse anti-LC3B, and rat anti-LAMP2a. The secondary antibodies were Alexa Fluor 488 donkey-anti-mouse (Thermo Fisher Scientific: A21202, 1/500), Alexa Fluor 568 donkey-anti-rabbit (Thermo Fisher Scientific: A10042, 1/500), and Alexa Fluor 488 goat-anti-rat (Thermo Fisher Scientific: A11006, 1/500). DAPI (ProLong Gold Antifade Reagent with DAPI; Life Technologies: P36935) was used to detect nuclei. Micrographs were acquired using a Nikon Eclipse Ti inverted microscope (Nikon) equipped with a Yokogawa CSU-X1 spinning disk unit (Yokagawa) and an Andor EMMCD camera (Andor Technology) using a 100x oil immersion objective lens. The co-localization dots were quantified by counting the number of LC3B-FTH1- or LAMP2a-FTH1-positive dots in 10 different areas (magnification 1000x) per section.

## Measurement of hematological parameters, serum ferritin, serum iron, transferrin saturation, serum troponin T, and serum IL-6

Blood samples were obtained from the inferior vena cava in anesthetized mice. Full blood count and reticulocyte count were measured at Pinmoore Animal Laboratory Services Limited. Blood samples were centrifuged for 30 min at 850 x g to isolate serum fraction. Serum ferritin levels were measured using a Mouse Ferritin ELISA Kit (FTL) (Abcam, ab157713) according to the manufacturer's protocols. Serum iron levels and transferrin saturation were measured using Pointe Scientific Iron/TIBC Reagents (Pointe Scientific, 23-666-320) according to the manufacturer's protocols. Serum troponin T levels were measured using the ELISA Kit for Troponin T Type 2, Cardiac (TNNT2) (Cloud-Clone, SED232Mu) according to the manufacturer's protocols. Serum IL-6 levels were measured using the ELISA Kit for Mouse IL-6 (R and D Systems, M6000B) according to the manufacturer's protocols.

## Measurement of total non-heme, ferrous, and ferric iron levels in hearts

Total non-heme, ferrous, and ferric iron in hearts or liver were analyzed using an Iron Assay Kit (Abcam, ab83366) according to the manufacturer's protocols. Briefly, the whole heart was perfused with saline and 10–20 mg of left ventricle tissue was homogenized in Iron Assay Buffer. The supernatant without the insoluble fraction was separated by centrifugation and used for analysis. A microplate reader was used to measure the absorbance at OD 593 nm. The level of ferric iron was calculated by subtracting ferrous iron from total non-heme iron. The ratio of the ferrous iron level to the FTH1 protein level was calculated to estimate non-binding fraction of ferrous iron to FTH1. The FTH1 protein level for sham-operated $Ncoa4^{+/+}$ estimated as Western blot analysis in hearts was set to 1.

## Measurement of malondialdehyde (MDA) in hearts

The amount of MDA in the hearts was measured using a Lipid Peroxidation (MDA) Assay Kit (Abcam, ab118970) according to the manufacturer's protocols. Briefly, 10–20 mg of fresh left ventricle tissue was homogenized in Lysis Solution containing butylated hydroxytoluene. The insoluble fraction was removed by centrifugation, and the supernatant was used for analysis. The supernatants were mixed with thiobarbituric acid (TBA) solution reconstituted in glacial acetic acid and then incubated at 95°C for 60 min. The supernatants containing MDA-TBA adduct were added into a 96-well microplate for analysis. A microplate reader was used to measure the absorbance at OD 532 nm.

## Glutathione quantification

Oxidized glutathione (GSSG) and total glutathione in hearts were analyzed using a GSSG/GSH Quantification Kit (Dojindo, G257) according to the manufacturer's protocols. Briefly, 20–30 mg of fresh left ventricle tissue was homogenized in 5% 5-sulfosalicylic acid (SSA), and the insoluble fraction was removed by centrifugation. The resultant supernatant was added to double-deionized $H_2O$ ($ddH_2O$) to reduce the SSA concentration to 0.5% for the assay. A microplate reader was used to measure absorbance at OD 415 nm. The concentration of reduced glutathione (GSH) was calculated by subtracting 2x GSSG from the total glutathione concentration.

## Measurement of glutamine and glutamate concentration in hearts

The glutamine concentration in hearts was analyzed using a Glutamine Assay Kit (Abcam, ab197011) according to the manufacturer's protocols. Briefly, 10–20 mg of fresh left ventricle tissue was

homogenized in ice-cold Hydrolysis Buffer, and the insoluble fraction was removed by centrifugation. The supernatant was added to perchloric acid (PCA). After 5 minutes incubation on ice, the samples were centrifuged and the supernatants were transferred into new tubes. To remove excess PCA, potassium hydroxide was added to the supernatant and the precipitated PCA was removed by centrifugation. A microplate reader was used to measure the absorbance at OD 450 nm. The glutamate concentration in hearts was analyzed using a Glutamate Assay Kit (Abcam, ab83389) according to the manufacturer's protocols. Briefly, 10–20 mg of fresh left ventricle tissue was homogenized in ice-cold Assay Buffer, and the insoluble fraction was removed by centrifugation. The supernatants were added into a 96-well microplate for analysis. A microplate reader was used to measure the absorbance at OD 450 nm.

## Free amino acid analysis by high–performance liquid chromatography (HPLC)

The mouse hearts were grinded with liquid nitrogen. Proteins were precipitated out from the mouse hearts using a 5% SSA solution, filtered and then measured by ion exchange chromatography with post column ninhydrin derivatization using a Biochrom 30+ amino acid analyzer with a lithium buffer system (Biochrom). EZChrom Elite software (Version 3.3.2.) was used for analysis.

## Electrophoretic mobility–shift assay (EMSA)

Electrophoretic mobility–shift assay (EMSA) was performed using Electrophoretic Mobility-Shift Assay kit (Invitrogen: E33075), according to the manufacturer's instructions. The following IRE containing mouse *Slc40a1* 5' UTR with T7 promoter was synthesized by Integrated DNA Technologies (gBlocks Gene Fragments): 5'-TAATACGACTCACTATAGGGGAGAGCAGGCTCGGGGTCTCC TGCGGCCGGTGGATCCTCCAACCCGCTCCCATAAGGCTTTGGCTTTCCAACTTCAGCTACAGTG TTAGCTAAGTTTGGAAAGAAGACAAAAAGAAGACCCCGTGACAGCTTTGCTGTTGTTGTTTGCC TTAGTTGTCCTTTGGGGTCTTTCGGCATAAGGCTGTTGTGCTTATACTGGTGCTATCTTCGGTTCC TCTCACTCCTGTGAACAAGCTCCCGGGCAAGAGCAGCTAAAGCTACCAGCAT-3'. The 287 bp fragment was cloned into pCR-Blunt II-TOPO (Invitrogen: 451245). This plasmid DNA containing the mouse *Slc40a1* 5' UTR was linearized by EcoRI and transcribed using HiScrib T7 Quick High Yield RNA Synthesis Kit (New England Biolabs: E2050S). Twenty micrograms of total protein homogenates from mouse heart were incubated with 50 ng of RNA oligonucleotides and subjected to electrophoresis on 6% nondenaturing polyacrylamide gels. The gels were stained using SYBR Green EMSA stain and captured using ChemiDoc-It Imaging Systems with Transilluminator (UVP).

## Isolation of mouse adult cardiomyocytes

Adult cardiomyocytes were isolated from 8- to 12-week-old male mice using a Langendorff system and cultured (*Oka et al., 2012*). Briefly, after $Ncoa4^{+/+}$ or $Ncoa4^{-/-}$ male mice had been deeply anesthetized, the heart was quickly excised, cannulated via the aorta, and perfused at constant flow. Hearts were first perfused for 1 minute at 37°C with a perfusion buffer containing 120 mM NaCl, 5.4 mM KCl, 1.6 mM $MgCl_2$, 1.2 mM $NaH_2PO_4$, 5.6 mM glucose, 20 mM $NaHCO_3$ and 5 mM taurine (Sigma-Aldrich), followed by collagenase buffer containing 1.2 mg/ml collagenase type 2 (Worthington Biochemical Corporation), and 0.016 mg/ml protease type XIV (Sigma-Aldrich: P-5147). After collagenase and protease digestion, the supernatant containing the dispersed myocytes was filtered into a sterilized tube and gently centrifuged at 20 x g for 3 minutes. The cell pellet was then promptly resuspended in perfusion buffer containing 200 µM $Ca^{2+}$. The cardiomyocytes were pelleted by gravity for 10 min, the supernatant was aspirated, and the cardiomyocytes were resuspended in perfusion buffer containing 500 µM $Ca^{2+}$. The final cell pellet was suspended in perfusion buffer containing 1 mM $Ca^{2+}$, and an appropriate amount of rod-shaped cardiomyocytes was then suspended in Minimum Essential Medium Eagle (MEM) (Sigma-Aldrich: M5650) supplemented with 2.5% fetal bovine serum, 2 mM L-glutamine, 100 U/ml penicillin, and 100 g/ml streptomycin (Sigma-Aldrich: G6784) and plated onto laminin (Invitrogen: 23017–015)-coated plates. After one hour of incubation in the culture medium, the cardiomyocytes were cultured in MEM (glutamine- and phenol red-free, Gibco: 51200038) supplemented with 1x MEM non-essential amino acids solution (Gibco: 11140035), 100 µg/ml bovine serum albumin, insulin (10 mg/l)-transferrin (5.5 mg/l)-sodium selenite

(6.7 µg/l) media supplement (ITS; Gibco: 41400045), 2 mM L-glutamine, 100 U/ml penicillin, and 100 g/ml streptomycin (Sigma-Aldrich: G6784).

## Cell death and ROS production

Cardiomyocyte death was estimated using a Live/Dead Viability/Cytotoxicity Assay Kit (Invitrogen). Isolated adult cardiomyocytes were pre-treated with or without 10 µM ferrostatin-1 (Sigma Aldrich: SML0583) for 30 min before treatment with 10 or 20 µM erastin (Sigma Aldrich: E7781), 10 or 100 µM isoproterenol (Sigma Aldrich: I5627), or 2 or 5 µM RSL3 (Sigma Aldrich: SML2234). The cells were then stimulated with or without ferrostatin-1 in the medium for four hours. After stimulation, the cells were stained with 1 mM calcein-AM (Invitrogen: C1430) and 2 µM ethidium homodimer-1 (Invitrogen: E1169) in the medium at 37°C for 10 min. The cells were washed three times using the medium and observed under a microscope (BZ-X700, Keyence). ROS production was measured by applying several indicators. After treatment with erastin or isoproterenol with or without ferrostatin-1, cells were stained with 25 µM 2', 7'-dichlorodihydrofluorescein diacetate (H2DCFDA; Invitrogen: D399) or C11-BODIPY (Invitrogen: D3861) in medium at 37°C for 10 min. The cells were then washed with the medium. ROS production was quantified using a fluorescence microplate reader.

## Measurement of the labile iron pool

The labile iron pool in the isolated adult cardiomyocytes was measured by the calcein-AM method (*Yoshida et al., 2019*). After treatment with indicated concentration of erastin or isoproterenol with or without 30 min treatment of 10 µM ferrostatin-1, cells were incubated with 1 µM calcein-AM at 37°C for 10 min and then washed three times with the medium. The fluorescence was measured using a fluorescence microplate reader. Then, the cells were treated with 10 µM pyridoxal isonicotinoyl hydrazine (PIH; Abcam: ab145871) at 37°C for 10 min, and washed three times with the medium. The fluorescence was measured again in a fluorescence microplate reader. The changes in fluorescence ($\Delta F$) upon PIH treatment was calculated for each sample.

## Administration of ferrostatin-1

Twenty-five mg of ferrostatin-1 was dissolved in 2.5 mL of DMSO, and then diluted with saline to the intended concentration. The final DMSO concentration was 5%. One day before and after TAC operation, the C57BL/6J mice or $Ncoa4^{flox/flox}$;Myh6-Cre$^+$ mice received an intraperitoneal injection of one mg/kg body weight ferrostatin-1 or saline containing 5% DMSO and every day thereafter. Following saline or ferrostain-1 injection, the mice were randomly assigned into sham and TAC groups.

## Statistics

The results are shown as the mean ± SEM. Statistical analyses were performed using GraphPad Prism 8 (GraphPad Software). Paired data were evaluated by unpaired, two-tailed Student's *t*-test. A one-way analysis of variance (ANOVA) followed by Tukey–Kramer's post hoc test was used for multiple comparisons. A two-way ANOVA followed by Tukey's multiple comparisons test was used for the in vitro experiments. $p < 0.05$ was considered to be statistically significant.

## Acknowledgements

We thank Dr Erika Cadoni, Dr Saki Nakagawa, and Mr. Darran Hardy for their excellent technical assistance. We thank Dr George Chennell and Ms Chen Liang, the Wohl Cellular Imaging Centre at King's College London, for help with spinning disk confocal microscopy. This work was supported by grants from the British Heart Foundation (CH/11/3/29051 and RG/16/15/32294), the Fondation Leducq (RA15CVD04), the European Research Council (692659), Japan Society for the Promotion of Science KAKENHI (18H02807), and Osaka University (International Joint Research Promotion Program) to K Otsu and from the British Heart Foundation to AM Shah (CH/1999001/11735 and RE/13/2/30182).

## Additional information

### Funding

| Funder | Grant reference number | Author |
|---|---|---|
| British Heart Foundation | CH/11/3/29051 | Kinya Otsu |
| British Heart Foundation | RG/16/15/32294 | Kinya Otsu |
| Fondation Leducq | RA15CVD04 | Kinya Otsu |
| H2020 European Research Council | 692659 | Kinya Otsu |
| Japan Society for the Promotion of Science | 18H02807 | Kinya Otsu |
| Osaka University | International Joint Research Promotion Program | Kinya Otsu |
| British Heart Foundation | CH/1999001/11735 | Ajay M Shah |
| British Heart Foundation | RE/13/2/30182 | Ajay M Shah |

The funders had no role in study design, data collection and interpretation, or the decision to submit the work for publication.

### Author contributions

Jumpei Ito, Conceptualization, Data curation, Software, Formal analysis, Validation, Investigation, Visualization, Methodology, Writing - original draft, Writing - review and editing; Shigemiki Omiya, Conceptualization, Data curation, Supervision, Validation, Investigation, Methodology, Writing - original draft, Writing - review and editing; Mara-Camelia Rusu, Data curation, Formal analysis, Validation, Investigation, Visualization, Methodology, Writing - review and editing; Hiromichi Ueda, Kazuki Nakahara, Data curation, Formal analysis, Validation, Investigation, Writing - review and editing; Tomokazu Murakawa, Yohei Tanada, Hajime Abe, Data curation, Formal analysis, Validation, Investigation, Methodology, Writing - review and editing; Michio Asahi, Supervision, Writing - review and editing; Manabu Taneike, Data curation, Formal analysis, Supervision, Validation, Investigation, Methodology, Writing - review and editing; Kazuhiko Nishida, Conceptualization, Resources, Software, Formal analysis, Supervision, Validation, Investigation, Visualization, Methodology, Writing - review and editing; Ajay M Shah, Resources, Funding acquisition, Validation, Methodology, Writing - review and editing; Kinya Otsu, Conceptualization, Supervision, Funding acquisition, Validation, Writing - original draft, Project administration, Writing - review and editing

### Author ORCIDs

Jumpei Ito (iD) https://orcid.org/0000-0001-5149-9962
Mara-Camelia Rusu (iD) http://orcid.org/0000-0002-8369-3580
Ajay M Shah (iD) https://orcid.org/0000-0002-6547-0631
Kinya Otsu (iD) https://orcid.org/0000-0001-9697-0711

### Ethics

Animal experimentation: All procedures were carried out in accordance with the King's College London Ethical Review Process Committee and the UK Home Office (Project License No. PPL70/8889) and were performed in accordance with the Guidance on the Operation of the Animals (Scientific Procedures) Act, 1986 (UK Home Office).

### Decision letter and Author response

Decision letter https://doi.org/10.7554/eLife.62174.sa1
Author response https://doi.org/10.7554/eLife.62174.sa2

## Additional files

### Supplementary files

• Supplementary file 1. Quantification and statistical analysis. The number of independent biological repeats (*n*) is shown in the figure legends. *P* values are shown below.

• Transparent reporting form

### Data availability

All data generated or analyzed during this study are included in the manuscript and supporting files.

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
