## [Decision Letter]

**Acceptance summary:**

This manuscript identified a role of iron-dependent cardiomyocyte death caused by autophagy-mediated ferritin degradation as a novel mechanism underlying chronic heart failure. The role of autophagy-dependent ferritin degradation was further strengthened by studies investigating nuclear receptor coactivator 4 (NCOA4), a cargo receptor that binds to the ferritin heavy chain. Its role was confirmed in an experimental model of chronic heart failure induced by transverse aortic constriction in mice carrying a cardiomyocyte-specific deletion of the NCOA4 gene.

**Decision letter after peer review:**

Thank you for submitting your article "Iron derived from autophagy-mediated ferritin degradation induces cardiomyocyte death and heart failure in mice" for consideration by *eLife*. Your article has been reviewed by three peer reviewers, one of whom is a member of our Board of Reviewing Editors, and the evaluation has been overseen by a Senior Editor. The reviewers have opted to remain anonymous.

The reviewers have discussed the reviews with one another and the Reviewing Editor has drafted this decision to help you prepare a revised submission.

Summary:

The manuscript describes the role of ferritinophagy in pressure overload induced heart failure. It demonstrates that deletion of the ferritin chaperone NCOA4, which is involved in ferritinophagy, reduces the extent of cardiac remodelling following TAC. It also provides evidence implicating changes in intracellular iron pool as a key driving factor in cardiac remodelling. Ferroptosis, a form of iron-induced cell death, has already been shown to be important in DOX-induced cardiomyopathy (Fang et al., 2019). The present study provides a further example of the importance of iron-induced cell death in heart failure. Overall, the study is well-designed and well presented. There are a number of gaps and inconsistencies in the data, and these should be addressed in order to achieve a water-tight and cohesive manuscript.

Essential revisions:

1) Prolonged expression of the alpha-MHC-Cre transgene used to target the heart in the present study is well known to have off-target effects on heart function, independently of any floxed allele. The authors need to demonstrate that the differences in the response to TAC between NCOA4^-/-^ and NCO4^+/+^ animals are not due to the alpha-MHC-Cre transgene altering the heart's response to pressure overload stress. This should be achieved by comparing the outcome of TAC between wild type and those harbouring the alpha-MHC-Cre transgene.

2) Quantitation of ferritinophagy- Histology (Figure 3C) and western blot (Figure 3A) relating to ferritin levels appear to show different things. The Western blot appears to show that sham-operated NCOA4^+/+^ and NCOA4^-/-^ hearts have similar levels of ferritin, whereas FTH1 staining shows that sham-operated NCO4-/- hearts have higher levels of ferritin staining. Additionally, the Western blot shows that TAC did not affect the levels of ferritin in NCOA4^-/-^ mice, whereas histology appears to show a reduction in ferritin staining by TAC. The authors need to provide better quality histological data, ensuring images are acquired in a manner that allows direct comparison between different treatment conditions.

3) Figure 4—figure supplement 1. The authors report that pressure overload reduced serum ferritin levels, and state that this suggests "systemic iron deficiency". This statement is not supported by their data because serum iron and transferrin saturation are comparable across the board. The authors need to provide additional data to explain the reduction in ferritin levels, taking into account the fact that ferritin is both a marker of liver iron stores and an acute inflammatory protein. Measurement of liver iron levels and serum inflammatory markers (IL-6 and or CRP) would help in the interpretation of this result.

4) Figure 4. The authors state that their measurements of iron levels suggest "free ferrous iron overload in TAC-operated *Ncoa4*^+/+^ hearts". At the same time, they report that the levels of FPN and TFR1 proteins are not altered in these hearts. These two findings contradict each other. Fpn and TfR1 are regulated by IRPs, which sense the levels of intracellular iron. If these increase, IRPs lose their RNA-binding function, thereby removing both the translational suppression on the Fpn transcript (resulting in increased FPN translation), and the stabilisation of the TfR1 transcript (resulting in reduced TfR1 expression). The authors need to provide more detailed analysis of IRP activity, using a more comprehensive panel of IRP-regulated genes and/or RNA binding assays. Crucially, they need to determine whether the IRP system is impaired in this setting or whether its activation is not sufficient to mitigate against changes in the size of the labile iron pool.

5) In Figure 4, while the authors characterized the glutathione and glutamine metabolism in pressure overload-induced cardiac hypertrophy, they did not examine the cystine-glutamate antiporter (System Xc-) that is a key regulator for cystine update in cell survival against ferroptosis. They should assess the expression of SLC7A11 in System Xc-, and cystine level in the cell. The expression level of glutamate also needs to be examined to evaluate the function of System Xc- in this setting. It might affect the level of L-glutamine, which is converted into glutamate by glutaminase.

6) In Figure 5, they used a ferroptosis inducer, erastin that inhibits the activity of System Xc-, leading to the depletion of GSH. To characterize the effects of NCOA4 deficiency in iron-dependent necrosis, they should use another ferroptosis inducer, RSL3 that inhibits GPX4 directly.

7) Figure 6. The data presented here are interesting but do not directly support the results seen in NCOA4^-/-^ mice. To integrate these results into the rest of the study and to support their statements on the role of iron-dependent cell death downstream of NCOA4, the authors need to determine whether ferrostatin provides additional protection from cardiac remodelling in NCOA4^-/-^ mice. If it does, then the protective effect of ferrostatin cannot be attributed to mitigating he effects of ferritinophagy.

8) While they discussed possible mechanisms of isoproterenol induced iron-dependent necrosis, it is unclear what is a potential mechanism for that. They need to discuss it more with additional data or references. For example, Isoproterenol or erastin-induced ferroptosis mechanism had been well known such as NRF2/HO-1pathway and VDACs-induced mitochondrial dysfunction (as reviewed by Li et al., 2020). Does NCO4 have association with these pathways in ferroptosis-induced cardiomyocytes death and heart failure?

9) Tang et al. (Free Radic Biol Med 2019 Apr; 134: 445-457) mentioned in their study that NCOA4 has association with mitochondria iron-overload in cardiomyocyte hypertrophy pathophysiology. Please discuss on the possibility that NCO4 has association with these pathways in ferroptosis-induced cardiomyocytes death and heart failure.

---

## [Author Response]

Essential revisions:1) Prolonged expression of the alpha-MHC-Cre transgene used to target the heart in the present study is well known to have off-target effects on heart function, independently of any floxed allele. The authors need to demonstrate that the differences in the response to TAC between NCOA4^-/-^ and NCO4^+/+^ animals are not due to the alpha-MHC-Cre transgene altering the heart's response to pressure overload stress. This should be achieved by comparing the outcome of TAC between wild type and those harbouring the alpha-MHC-Cre transgene.

I agree with your comments that prolonged expression of the alpha-MHC-Cre transgene in some transgenic lines is known to have off-target effects on heart function as previously reported (J Card Fail. 2006, 12:392-398). Our alpha-MHC-Cre (Myh6-Cre) transgenic mice have been generated in our laboratory (J Clin Invest 2004, 114:937-843) and used for a series of our and others’ studies. In this study, please note that control *Ncoa4*^flox/flox^;Myh6-Cre^–^ mice developed more severe chamber dilation and cardiac dysfunction than *Ncoa4*^flox/flox^;Myh6-Cre^+^ mice. Thus, the reported detrimental effect of prolonged expression of the transgene on cardiac function cannot explain our data showing ablation of *Ncoa4* induced cardioprotective effects. However, in the revised manuscript, we performed TAC operation on Myh6-Cre^+^ and Myh6-Cre^–^ mice. There were no significant differences in echocardiographic parameters between the two groups 4 weeks after TAC. Thus, the overexpression of Cre recombinase in the heart has no effect on pressure overload-induced cardiac remodeling. We showed the data in Figure 1—figure supplement 4 and mentioned the data in the Results section.

2) Quantitation of ferritinophagy- Histology (Figure 3C) and western blot (Figure 3A) relating to ferritin levels appear to show different things. The Western blot appears to show that sham-operated NCOA4^+/+^ and NCOA4^-/-^ hearts have similar levels of ferritin, whereas FTH1 staining shows that sham-operated NCO4^-/-^ hearts have higher levels of ferritin staining. Additionally, the Western blot shows that TAC did not affect the levels of ferritin in NCOA4^-/-^ mice, whereas histology appears to show a reduction in ferritin staining by TAC. The authors need to provide better quality histological data, ensuring images are acquired in a manner that allows direct comparison between different treatment conditions.

Thank you very much for your comments. We have provided better quality of histological data (Figure 3C).

3) Figure 4—figure supplement 1. The authors report that pressure overload reduced serum ferritin levels, and state that this suggests "systemic iron deficiency". This statement is not supported by their data because serum iron and transferrin saturation are comparable across the board. The authors need to provide additional data to explain the reduction in ferritin levels, taking into account the fact that ferritin is both a marker of liver iron stores and an acute inflammatory protein. Measurement of liver iron levels and serum inflammatory markers (IL-6 and or CRP) would help in the interpretation of this result.

Thank you very much for your comments. In the revised manuscript, we showed the liver iron and serum IL-6 levels in Figure 4—figure supplement 2. TAC decreased the level of total non-heme iron in both *Ncoa4^-/-^* and *Ncoa4^+/+^* livers, and there was no significant difference in the level between *Ncoa4^-/-^* and *Ncoa4^+/+^* livers. TAC increased the serum IL-6 level in *Ncoa4^-/-^* and *Ncoa4^+/+^* mice, which was higher in *Ncoa4^+/+^* than that in *Ncoa4^-/-^* mice. It has been reported that serum IL-6 level is upregulated in heart failure. Taking into account the fact that ferritin is both a marker of liver iron stores and an acute inflammatory protein as you indicated, the reduced level of serum ferritin in pressure overloaded hearts is due to a reduction in liver iron stores. We deleted the description “systemic iron deficiency” in the Results and Discussion sections and mentioned the data and interpretation in the Results section.

4) Figure 4. The authors state that their measurements of iron levels suggest "free ferrous iron overload in TAC-operated Ncoa4^+/+^ hearts". At the same time, they report that the levels of FPN and TFR1 proteins are not altered in these hearts. These two findings contradict each other. Fpn and TfR1 are regulated by IRPs, which sense the levels of intracellular iron. If these increase, IRPs lose their RNA-binding function, thereby removing both the translational suppression on the Fpn transcript (resulting in increased FPN translation), and the stabilisation of the TfR1 transcript (resulting in reduced TfR1 expression). The authors need to provide more detailed analysis of IRP activity, using a more comprehensive panel of IRP-regulated genes and/or RNA binding assays. Crucially, they need to determine whether the IRP system is impaired in this setting or whether its activation is not sufficient to mitigate against changes in the size of the labile iron pool.

We have analyzed of IRP activity by examining mRNA levels of IRP-regulated genes such as *Tfrc*, *Slc11a2*, *Cdc14a*, and *Cdc42bpa* (Figure 4—figure supplement 3B) and RNA binding activity for ferroportin 1 (SLC40A1) (Figure 4—figure supplement 3C). There were no significant differences in all parameters between TAC-operated *Ncoa4^-/-^* and *Ncoa4^+/+^* mice. Thus, the data indicates that the IRP system is impaired in this setting. We mentioned this in the Results section.

5) In Figure 4, while the authors characterized the glutathione and glutamine metabolism in pressure overload-induced cardiac hypertrophy, they did not examine the cystine-glutamate antiporter (System Xc-) that is a key regulator for cystine update in cell survival against ferroptosis. They should assess the expression of SLC7A11 in System Xc-, and cystine level in the cell. The expression level of glutamate also needs to be examined to evaluate the function of System Xc- in this setting. It might affect the level of L-glutamine, which is converted into glutamate by glutaminase.

We examined the expression level of *Slc7a11* mRNA. There was no significant difference between any groups (Figure 4—figure supplement 5A). Cardiac cystine level showed no difference between TAC-operated *Ncoa4^-/-^* and *Ncoa4^+/+^* mice (Figure 4—figure supplement 5B). The level of glutamate in TAC-operated *Ncoa4^+/+^* hearts was lower than that in sham-operated *Ncoa4^+/+^* or TAC-operated *Ncoa4^-/-^* hearts (Figure 4—figure supplement 5C). We described the data in the Results section and the Discussion section.

6) In Figure 5, they used a ferroptosis inducer, erastin that inhibits the activity of System Xc-, leading to the depletion of GSH. To characterize the effects of NCOA4 deficiency in iron-dependent necrosis, they should use another ferroptosis inducer, RSL3 that inhibits GPX4 directly.

In the revised manuscript, we examined the effect of RSL3 on cardiomyocyte death. RSL3 induced cardiomyocyte death, which was attenuated by *Ncoa4* ablation or ferrostatin-1 treatment as erastin (Figure 5—figure supplement 2). We mentioned the data in the Results section.

7) Figure 6. The data presented here are interesting but do not directly support the results seen in NCOA4^-/-^ mice. To integrate these results into the rest of the study and to support their statements on the role of iron-dependent cell death downstream of NCOA4, the authors need to determine whether ferrostatin provides additional protection from cardiac remodelling in NCOA4^-/-^ mice. If it does, then the protective effect of ferrostatin cannot be attributed to mitigating he effects of ferritinophagy.

In the revised manuscript, we examined the effect of ferrostatin-1 on pressure overload-induced cardiac remodeling in *Ncoa4^-/-^* mice. Four weeks after TAC, there were no significant differences in echocardiographic and physiological parameters between saline- and ferrostatin-1-treated mice (Figure 6—figure supplement 2). Thus, this result indicates that iron-dependent cell death is downstream of NCOA4-mediated pathway. We mentioned these in the Results and Discussion sections.

8) While they discussed possible mechanisms of isoproterenol induced iron-dependent necrosis, it is unclear what is a potential mechanism for that. They need to discuss it more with additional data or references. For example, Isoproterenol or erastin-induced ferroptosis mechanism had been well known such as NRF2/HO-1pathway and VDACs-induced mitochondrial dysfunction (as reviewed by Li et al., 2020). Does NCO4 have association with these pathways in ferroptosis-induced cardiomyocytes death and heart failure?

In the revised manuscript, we examined the effect of isoproterenol on ferritinophagy. Isoproterenol decreased the FTH1 protein level in isolated cardiomyocytes in an NCOA4 dependent manner, indicating that isoproterenol induces cardiomyocyte death by activation of NCOA4-mediated ferritinophagy (Figure 5E). We mentioned these in the Results and Discussion sections.

We added the sentence “It is possible that NCOA4 may associate with NRF2/HO-1 and VDACs-induced mitochondrial dysfunction pathways in ferroptosis-induced cardiomyocytes death and heart failure” in the Discussion section.

9) Tang et al. (Free Radic Biol Med 2019 Apr; 134: 445-457) mentioned in their study that NCOA4 has association with mitochondria iron-overload in cardiomyocyte hypertrophy pathophysiology. Please discuss on the possibility that NCO4 has association with these pathways in ferroptosis-induced cardiomyocytes death and heart failure.

Thank you very much. In the revised manuscript, we mentioned mitochondria iron-overload as a possible mechanism in cardiomyocyte hypertrophy pathophysiology in the Discussion section. Although NCOA4-mediated ferritin degradation contributes to maintain mitochondrial function through iron supply, excessive ferritinophagy may induce cardiomyocyte hypertrophy and cell death. We need further study to elucidate the mechanism for NCOA4-medicated cardiomyocyte death and heart failure.